# Stress-induced Cdk5 activity enhances cytoprotective basal autophagy in *Drosophila melanogaster* by phosphorylating acinus at serine[437]

**Nilay Nandi[1], Lauren K Tyra[1], Drew Stenesen[1], Helmut Krämer[1,2]\***

[1]Department of Neuroscience, UT Southwestern Medical Center, Dallas, United States; [2]Department of Cell Biology, UT Southwestern Medical Center, Dallas, United States

**Abstract** Cdk5 is a post-mitotic kinase with complex roles in maintaining neuronal health. The various mechanisms by which Cdk5 inhibits and promotes neurodegeneration are still poorly understood. Here, we show that in *Drosophila melanogaster* Cdk5 regulates basal autophagy, a key mechanism suppressing neurodegeneration. In a targeted screen, Cdk5 genetically interacted with Acinus (Acn), a primarily nuclear protein, which promotes starvation-independent, basal autophagy. Loss of Cdk5, or its required cofactor p35, reduces S437-Acn phosphorylation, whereas Cdk5 gain-of-function increases pS437-Acn levels. The phospho-mimetic S437D mutation stabilizes Acn and promotes basal autophagy. In *p35* mutants, basal autophagy and lifespan are reduced, but restored to near wild-type levels in the presence of stabilized Acn$^{S437D}$. Expression of aggregation-prone polyQ-containing proteins or the Amyloid-β42 peptide, but not alpha-Synuclein, enhances Cdk5-dependent phosphorylation of S437-Acn. Our data indicate that Cdk5 is required to maintain the protective role of basal autophagy in the initial responses to a subset of neurodegenerative challenges.
DOI: https://doi.org/10.7554/eLife.30760.001

**\*For correspondence:**
helmut.kramer@utsouthwestern.edu

**Competing interests:** The authors declare that no competing interests exist.

## Introduction

Cdk5 shares strong homology with other members of the family of cyclin-dependent kinases (Cdks), but it is distinct in its modes of regulation and function (*Dhavan and Tsai, 2001*; *Pozo and Bibb, 2016*). Unlike other Cdks, Cdk5 is best known for its function in post-mitotic cells rather than cell cycle regulation (*Dhariwala and Rajadhyaksha, 2008*). In post-mitotic cells, Cdk5 is regulated by binding to its obligatory membrane-associated p35 or p39 co-activators (*Tsai et al., 1994*; *Tang et al., 1995*). These co-activators are highly expressed in the brain and loss of Cdk5 activity in mice or flies has been associated with defects in neurite outgrowth (*Su and Tsai, 2011*; *Trunova et al., 2011*), neuronal migration (*Nishimura et al., 2014*), pre- and post-synaptic functions (*Bibb et al., 1999*; *Li et al., 2001*), the maintenance of synaptic plasticity (*Hawasli et al., 2007*), retinal degeneration (*Kang et al., 2012*), and neurodegeneration in aging brains (*Trunova and Giniger, 2012*; *Shah and Lahiri, 2017*). Accordingly, dysregulation of Cdk5 has been observed in numerous brain diseases, including schizophrenia and epilepsy (*Patel et al., 2004*; *Engmann et al., 2011*) and neurodegenerative disorders, including Huntington's disease, Alzheimer's disease and Amyotrophic Lateral Sclerosis (ALS) (*Cheung and Ip, 2012*). Cdk5 substrates regulating microtubule-based transport (*Klinman and Holzbaur, 2015*) and synaptic function (*Tan et al., 2003*; *Lai and Ip, 2015*), have been identified, but the long-term neuroprotective function of Cdk5 activity remains poorly understood (*McLinden et al., 2012*; *Meyer et al., 2014*).

**eLife digest** Cells have a problem that we recognize from our own homes: if nobody cleans up, garbage accumulates. Unwanted material in cells can include proteins that clump together and can no longer carry out their normal tasks. If left to build up, these protein aggregates can damage the cell and even kill it. Many neurodegenerative disorders, including Huntington's disease and Alzheimer's disease, arise when such faulty proteins accumulate inside brain cells.

Autophagy is a process that can destroy protein aggregates and other defective material to keep cells healthy. Understanding how cells regulate autophagy is thus of great interest to scientists. A protein called Acinus promotes autophagy and is found in many organisms including fruit flies and humans. All Acinus proteins share a common feature; they contain a site called Serine[437] that can be modified by the attachment of a phosphate group, in a process known as phosphorylation. However, the significance of this modification was not clear.

Nandi et al. have now asked if this phosphorylation event is important for the role of Acinus in autophagy. The experiments were carried out in the fruit fly, *Drosophila melanogaster*. Flies were engineered such that the normal Acinus protein was replaced with a mutant version that mimics the phosphorylation at Serine[437]. These mutant flies had higher levels of Acinus, showed more autophagy, and lived longer when compared to normal flies.

Further work identified a protein called Cdk5 as being responsible for attaching phosphate to Acinus at Serine[437]. Making Cdk5 inactive using experimental tools led to lower levels of autophagy in brain cells and shortened the flies' life span. Moreover, some aggregation-prone proteins linked to neurodegenerative diseases can enhance the activity of Cdk5 towards Acinus, thereby reducing their own accumulation through elevated autophagy.

Together these findings show that phosphorylation of Acinus by Cdk5 maintains healthy brain cells and improves life span by enhancing autophagy. The next step is to understand how phosphorylation at Serine[437] stabilizes Acinus to boost autophagy. This may lead to new ways to adjust the levels of autophagy to benefit different organisms.

DOI: https://doi.org/10.7554/eLife.30760.002

Autophagy, here short for macroautophagy, is a key cellular process for maintaining the health of neurons (*Menzies et al., 2015*). Cytoprotective functions of autophagosomes in neurons include the engulfment and lysosomal delivery of aggregates of misfolded proteins and the disposal of dysfunctional mitochondria (*Green and Levine, 2014*). This protective role of basal autophagy was demonstrated by neuron-specific mutations in autophagy core components: neuronal loss of Atg5 or Atg7 yields rapid neurodegeneration (*Hara et al., 2006*; *Komatsu et al., 2006*; *Juhász et al., 2007*). These findings are consistent with results in mouse and Drosophila models of Huntington's disease or spinocerebellar ataxia type 3 (SCA3) in which elevated autophagy corresponded to reduced loads of aggregated Huntingtin (Htt) protein and ameliorated neuronal phenotypes (*Ravikumar et al., 2004*; *Bilen and Bonini, 2007*; *Sarkar et al., 2007*; *Zheng et al., 2010*; *Jaiswal et al., 2012*; *Menzies et al., 2015*). Although the rapid induction of autophagy in response to glucose or amino-acid deprivation is well described (*Galluzzi et al., 2014*), little is known about the modulation of basal levels of autophagy in response to stress caused by protein aggregates (*Ashkenazi et al., 2017*).

We have previously identified Acinus (Acn) as a novel regulator of basal autophagy in *Drosophila* (*Haberman et al., 2010*; *Nandi et al., 2014*). Mammalian Acn had originally been identified as a caspase target aiding in chromatin modifications in apoptotic cells (*Sahara et al., 1999*; *Joselin et al., 2006*). In mammalian and *Drosophila cells*, Acn is highly enriched in the nucleus where, together with its binding partners Sap18 and RNPS1, it forms the ASAP complex (*Schwerk et al., 2003*; *Murachelli et al., 2012*). ASAP interacts with the exon junction complex (*Tange et al., 2005*) and participates in the regulation of alternative splicing (*Schwerk et al., 2003*; *Jang et al., 2008*; *Hayashi et al., 2014*; *Malone et al., 2014*). Antagonistic activities of Akt1 kinase and caspase-3 homologs regulate Acn levels (*Hu et al., 2005*) and genetic manipulations that prevent the caspase-mediated cleavage of endogenous Acn in *Drosophila* can elevate its levels in a cell type-specific manner (*Nandi et al., 2014*).

An unexpected consequence of such elevated Acn was an increase in basal, starvation-independent autophagy. High-level Acn overexpression by the Gal4/UAS system triggers autophagy-dependent death in *Drosophila* (*Haberman et al., 2010*). By contrast, Acn levels were modestly increased by the Acn[D527A] mutation that interferes with its Caspase-mediated cleavage or by phospho-mimetic mutations in two AKT1 target sites that reduce this cleavage (*Nandi et al., 2014*). Such mildly elevated Acn levels yielded elevated basal autophagy with beneficial outcomes including enhanced starvation resistance, prolonged life span and reduced loads of polyQ aggregates in a *Drosophila* model of Huntington's disease (*Nandi et al., 2014*).

Here, we show that similar benefits are gained by phosphorylation of Acn at serine 437. We identify Cdk5 as the kinase which mediates this phosphorylation and show that Cdk5 activity is enhanced in the presence of multiple aggregation-prone proteins, including Huntingtin-Q93 (Htt.Q93), SCA3.Q78, and Amyloid-beta peptide 42 (Aβ42). These findings offer new insights into the complex mechanisms balancing the effects of loss and gain-of-function of Cdk5 on neurodegenerative diseases.

## Results

### Phosphorylation of conserved serine 437 stabilizes Acn

Phosphorylation of conserved C-terminal Akt1-target sites (*Figure 1A*) regulates Acn levels in flies and mammalian cells (*Hu et al., 2005*; *Nandi et al., 2014*). This motivated us to investigate the functional consequences of phosphorylation of another highly conserved residue of Acn, namely serine 437 (*Figure 1B*). Phosphorylated S437 has been detected by phosphoproteomics approaches in mammalian cancer cells (*Patwa et al., 2008*; *Francavilla et al., 2017*) and *Drosophila* (*Bodenmiller et al., 2007*).

To explore phosphorylation of this residue in vivo, we raised an antibody that specifically recognizes Acn phosphorylated at serine 437. To assess its specificity for S437-phosphorylated Acn (pS437-Acn), we generated flies in which endogenous Acn was replaced by N-terminally Myc-tagged Acn[WT] or Acn[S437A] (*Figure 1A*). Expression of these transgenes was under control of the endogenous *acn* promoter and enhancers within a 4 kb *acn* genomic region (*Nandi et al., 2014*). All genomic *acn* transgenes were inserted at 96F3 to avoid insertion site-specific differences in expression levels (*Figure 1K*). The previously observed lethality, developmental defects and endocytic trafficking defects of *acn*-null alleles were rescued by both transgenes and also a corresponding phospho-mimetic Acn[S437D] (*Figure 1—figure supplement 1*). From here on, we will refer to these rescued flies that, in an *acn*-null background exclusively express transgenic forms of Acn as Acn[WT], Acn[S437D] or Acn[S437A] flies. Full genotypes for each experiment are listed in *Supplementary file 3*.

Staining of Acn[WT] larval eye discs with the phospho-specific pS437-Acn antibody revealed a dynamic expression pattern consistent with staining for the Myc epitope tag (*Figure 1C*) and the previously described distribution of Acn in retinal cells (*Haberman et al., 2010*; *Nandi et al., 2014*). Two approaches were used to assess specificity for pS437-Acn. First, staining with pS437-Acn antibody of Acn[S437A] eye discs was reduced to background level (*Figure 1D*). Second, treatment of Acn[WT] eye discs with Calf Intestinal Phosphatase (CIP) reduced staining for pS437-Acn but not the Myc epitope (*Figure 1E*), demonstrating the specificity of the pS437-Acn antibody and the high level of Acn phosphorylation at S437 in developing eye discs.

To test the effect of S437 phosphorylation on Acn levels, we stained the larval eye disc with Acn antibody. Acn levels were slightly reduced in Acn[S437A] eye discs compared to Acn[WT] (*Figure 1F,G*). The small difference between Acn[WT] and Acn[S437A] levels in western blots of larval lysates (*Figure 1I*) may reflect the contribution of other tissues with low levels of S437 phosphorylation of Acn[WT] or with Acn[S437A] stabilized by alternative mechanisms. By contrast, Acn levels were further enhanced in flies expressing only the phospho-mimetic Acn[S437D] (*Figure 1H*). This was consistent with changes in Acn levels when compared by western blot analysis of larval lysates: Acn[S437D] levels were elevated compared to Acn[WT] or phospho-inert Acn[S437A] (*Figure 1I,J*). Moreover, we detected a smaller form of Acn that lost the N-terminal Myc-epitope (arrow in *Figure 1I*). This likely represents a cleaved form of Acn that was stabilized for phospho-mimetic Acn[S437D] but degraded for Acn[WT] or Acn[S437A] (*Figure 1I*). Proteolytic cleavage of Acn close to the S437 residue has previously been observed for *Drosophila* and mammalian Acn proteins (*Sahara et al., 1999*; *Hu et al., 2005*; *Nandi et al., 2014*).

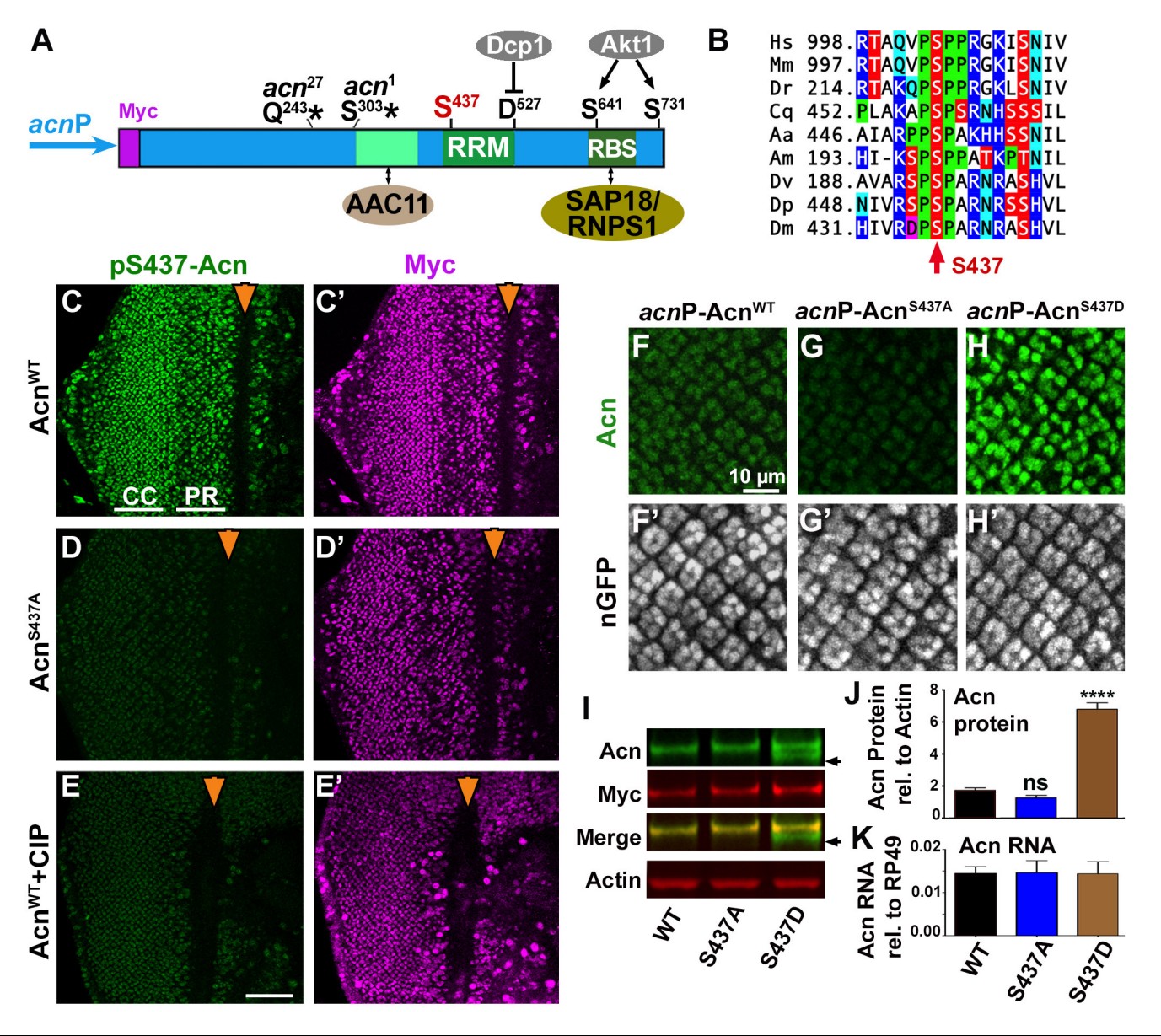

**Figure 1.** Acn protein is stabilized by phosphorylation at Serine 437. (A) Cartoon shows regulatory elements of Acn. Genomic Acn transgenes are Myc-tagged and expressed from the endogenous *acn*P promoter in the background of *acn*[1] and *acn*[27] null alleles. Green boxes indicate the RRM domain and regions that bind to AAC11 (*Rigou et al., 2009*) or SAP18/RNPS1 (*Murachelli et al., 2012*). Phosphorylation at the AKT1 target sites S641 and S731 reduces Dcp-1-mediated cleavage at D527 (*Nandi et al., 2014*). (B) Serine S437 is conserved from humans to insects. Sequences shown: *Homo sapiens* NP_055792.1; *Mus musculus* AAF89661.1; *Danio rerio* AAI16537.1; *Culex quinquefasciatus* XP_001846490.1; *Aedes aegypti* XP_001664312.1; *Apis mellifera* XP_006570961.1; *Drosophila virilis* XP_015028002.1; *Drosophila persimilis* XP_002014510.1; *Drosophila melanogaster* NP_609935.1. (C–E) Projections of confocal micrographs of eye discs from Acn[WT] (C,E) or Acn[S437A] (D) larvae stained for pS437-Acn and Myc. Strong staining for pS437-Acn is dominated by cone cells (CC) in posterior regions of eye discs, and by photoreceptors cells (PR) closer to the furrow (arrowhead). Staining for pS437-Acn, but not Myc, is reduced to background level in Acn[S437A] eye discs (D) or after calf intestinal phosphatase (CIP) treatment of Acn[WT] (E). (F–H) Micrographs focusing on early photoreceptor clusters in eye disc stained for Acn and DNA. Compared to Acn[WT] (F), Acn levels are reduced for Acn[S437A] and increased for the phosphomimetic Acn[S437D]. (I,J) Western blots of larval lysates show elevated levels of Acn[S437D] compared to Acn[WT] or Acn[S437A]. An Acn cleavage product (arrow) removing the N-terminal Myc tag is stable in Acn[S437D], but not Acn[WT] or Acn[S437A]. (J) Acn levels relative to the Actin loading control are quantified from three blots from biological repeats. (K) RT-qPCR reveals equal expression for Acn transgenes controlled by the endogenous *acn*P promoter relative to ribosomal protein RP49. Bar graphs show mean ±SD. ns, not significant; ****p<0.0001; each compared to wild-type Acn control. Scale bar in E is 50 μm in C-E, scale bar in F is 10 μm in F-H. Detailed genotypes are listed in *Supplementary file 3*.

DOI: https://doi.org/10.7554/eLife.30760.003

*Figure 1 continued on next page*

*Figure 1 continued*

The following source data and figure supplement are available for figure 1:

**Source data 1.** Quantification of Acn proteins in Western blots.

DOI: https://doi.org/10.7554/eLife.30760.005

**Figure supplement 1.** Acn$^{S437A}$ and Acn$^{S437D}$ support normal Sevenless-mediated Boss endocytosis.

DOI: https://doi.org/10.7554/eLife.30760.004

Together, these data indicate that Acn S437 is phosphorylated in developing tissues and plays a critical role in regulating Acn levels.

## Stabilized Acn elevates basal autophagy

As altering Acn levels can modulate the level of basal autophagy (*Nandi et al., 2014*), we tested whether the stabilized Acn$^{S437D}$ mutant exhibited elevated levels of autophagy. First, we analyzed endogenous Atg8a in eye discs of fed wandering third instar larvae. Atg8a punctae mark autophagosomes and early autolysosomal structures (*Klionsky et al., 2016*). The number of Atg8a-positive punctae was higher for phospho-mimetic Acn$^{S437D}$ eye discs compared to Acn$^{WT}$ or phospho-inert Acn$^{S437A}$, indicating elevated levels of autophagy (*Figure 2A–D*). Another tissue with highly regulated autophagy are larval fat bodies (*Rusten et al., 2004*; *Scott et al., 2004*). When we examined fat bodies of fed 96 hr larvae, Atg8a punctae were rare in Acn$^{WT}$ or Acn$^{S437A}$ larvae, but numerous and brightly stained in fed Acn$^{S437D}$ larval fat bodies (*Figure 2E–H*, *Figure 2—figure supplement 1A*). Importantly, a 4-hr amino acid starvation further increased the accumulation of Atg8a punctae in all three genotypes (*Figure 2I–L*, *Figure 2—figure supplement 1*).

We used three approaches to distinguish whether the elevated levels of Atg8a punctae in fed Acn$^{S437D}$ larvae represent an accumulation of stalled autophagosomes, or elevated flux through the pathway.

First, we inhibited lysosomal acidification and degradation with chloroquine to reveal Atg8a that has been delivered to autophagosomes and otherwise would be degraded (*Lőw et al., 2013*; *Mauvezin et al., 2014*). For Acn$^{WT}$, Acn$^{S437A}$ and Acn$^{S437D}$ fat bodies, chloroquine treatment resulted in further elevation of ATG8a staining after starvation, consistent with elevated flux in these starved tissues (*Figure 2I'–L*, *Figure 2—figure supplement 1*). Chloroquine treatment also significantly enhanced ATG8a staining in fed Acn$^{S437D}$ fat bodies consistent with elevated autophagic flux (*Figure 2G', H*, *Figure 2—figure supplement 1*).

Second, we used LysoTracker to evaluate acidification of autolysosomes and lysosomes (*DeVorkin and Gorski, 2014a*). Under fed conditions, few LysoTracker-positive punctae were detected in Acn$^{WT}$ or Acn$^{S437A}$ fat bodies, and their number increased upon amino acid starvation (*Figure 2M–P*), consistent with previous reports of starvation-induced autophagic flux in larval fat bodies (*Rusten et al., 2004*; *Scott et al., 2004*). Acn$^{S437D}$ larval fat bodies, however, displayed numerous LysoTracker-positive structures even in fed 96 hr larvae (*Figure 2O,P*) and their number further significantly increased after a 4 hr amino acid starvation (*Figure 2O',P*).

Third, we analyzed these changes on the ultrastructural level using transmission electron microscopy (TEM). Previous work has shown that fat bodies from fed wild-type larvae contain predominately lysosomes smaller than 400 μm, while starved fat bodies contain lysosomes and autolysosomes larger than 1 μm (e.g. *Scott et al., 2004*; *Takáts et al., 2013*; *Takáts et al., 2014*). When we analyzed fat bodies of fed 96 hr Acn$^{S437D}$ larvae, we observed significantly larger and more abundant lysosomes and autolysosomes compared to Acn$^{WT}$ (*Figure 2Q,R*). The mean diameter of lysosomal/autolysosomal structures is increased more than eight-fold from $250 \pm 20$ nm in Acn$^{WT}$ to $2044 +/-135$ nm in Acn$^{S437D}$ larvae (*Figure 2—figure supplement 2S*) and the mean area they occupied was increased almost 65-fold (*Figure 2T*).

We previously observed that increased levels of Acn exert a concentration-dependent physiological response. Mild elevation, for example by preventing caspases-mediated cleavage of Acn, enhanced starvation resistance and extended life span of well-fed flies (*Nandi et al., 2014*). By contrast, da-Gal4-driven expression at higher levels caused autophagy-mediated lethality (*Haberman et al., 2010*). Therefore, we explored the physiological consequences of elevated autophagy in Acn$^{S437D}$ flies. When challenged with starvation stress, Acn$^{S437D}$ flies survived

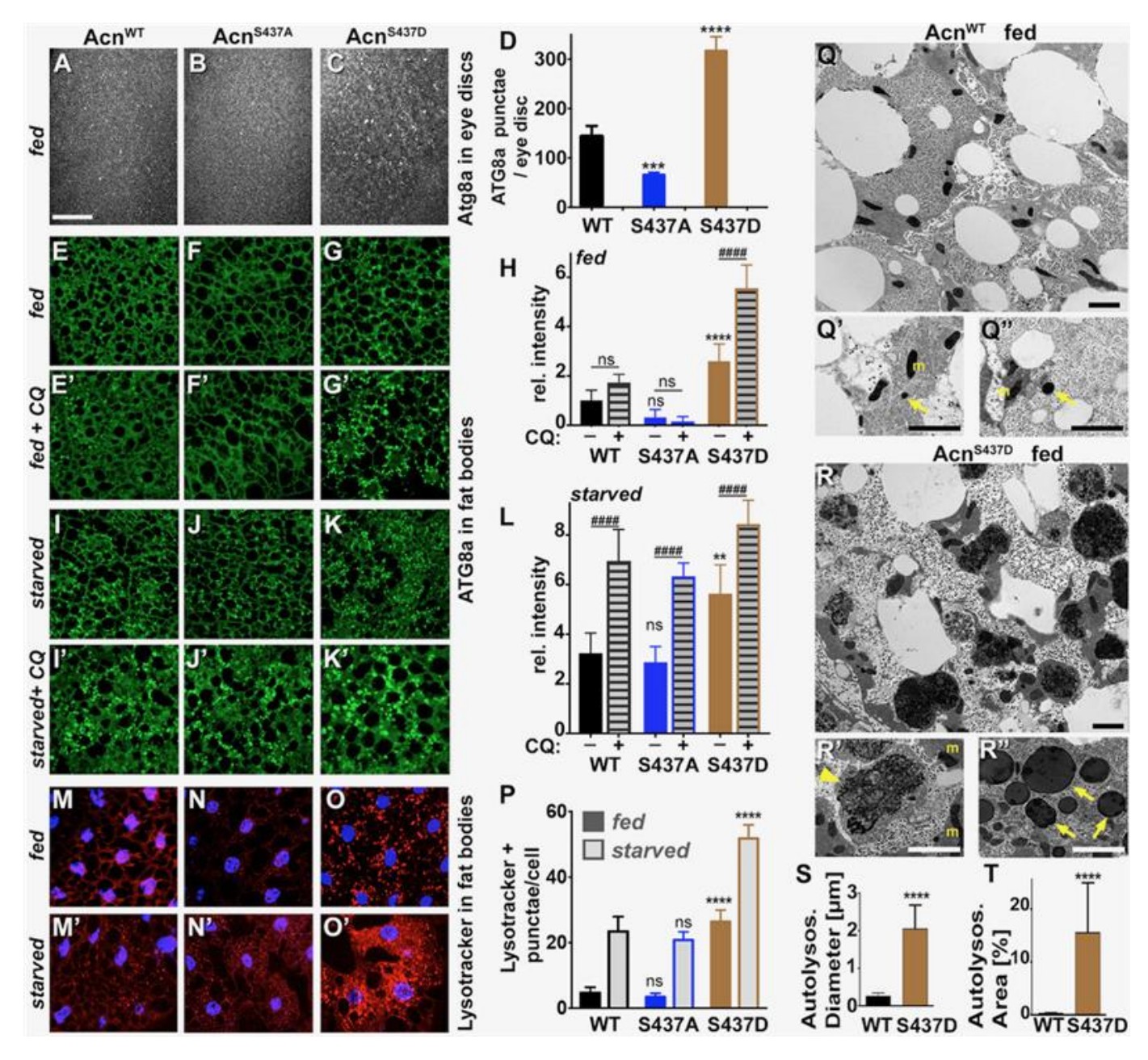

**Figure 2.** Stabilized Acn[S437D] enhances basal autophagy. (A–C) Micrographs of fed Acn[WT], Acn[S437A] or Acn[S437D] larval eye discs stained for Atg8a. Scale bar: 40 μm in A-O. (D) Quantification of Atg8a punctae in eye discs from five larvae. (E–G, I–K, M–O) Micrographs of Acn[WT], Acn[S437A] or Acn[S437D] larval fat bodies encompassing 6 to 8 cells from 96 hr fed or starved size-matched larvae (as indicated) stained for ATG8a (E–G, I–K) or with Lysotracker (M–O). For panels E'-G' and I'-K' lysosomal degradation was inhibited with chloroquine to visualize autophagic flux. (H,L,P) Quantification of Atg8a intensity (H,L) or Lysotracker punctae (P) in fat bodies averaged from six to eight cells from four to five larvae from one representative experiment out of three repeats. (Q,R) TEMs of fed Acn[WT] or Acn[S437D] fat bodies. Smaller panels show higher magnification examples of dense lysosomes (arrowheads), membrane-enriched autolysosomes (arrows) and mitochondria (m). Scale bars are 2 μm. (S) Quantification of diameters of at least 100 lysosomes and autolysosomes per genotype. (T) Quantification of percentage of autolysosomal area averaged from 25 images per genotype. Bar graphs show mean ± SD. **p<0.01; ****p<0.0001; ns, not significant; each compared to corresponding fed or starved wild-type Acn control. For each genotype, starved and fed were significantly different (p<0.01). Significant differences between untreated and chloroquine-treated larvae are indicated (##p<0.01; ###p<0.001, ####p<0.0001). Detailed genotypes are listed in *Supplementary file 3*.

DOI: https://doi.org/10.7554/eLife.30760.006

The following source data and figure supplement are available for figure 2:

*Figure 2 continued on next page*

*Figure 2 continued*

**Source data 1.** Relates to *Figure 2D*.
DOI: https://doi.org/10.7554/eLife.30760.008
**Source data 2.** Relates to *Figure 2H and L*.
DOI: https://doi.org/10.7554/eLife.30760.009
**Source data 3.** Relates to *Figure 2P*.
DOI: https://doi.org/10.7554/eLife.30760.010
**Source data 4.** Quantification of autolysosome and lysosome diameters and relative area.
DOI: https://doi.org/10.7554/eLife.30760.011
**Figure supplement 1.** Quantification of Atg8a punctae in fat bodies.
DOI: https://doi.org/10.7554/eLife.30760.007

significantly longer than Acn$^{WT}$ or Acn$^{S437D}$ flies (*Figure 3A*). Furthermore, Acn$^{S437D}$ life span under well-fed conditions was significantly extended, whereas the phospho-inert Acn$^{S437A}$ mutants died

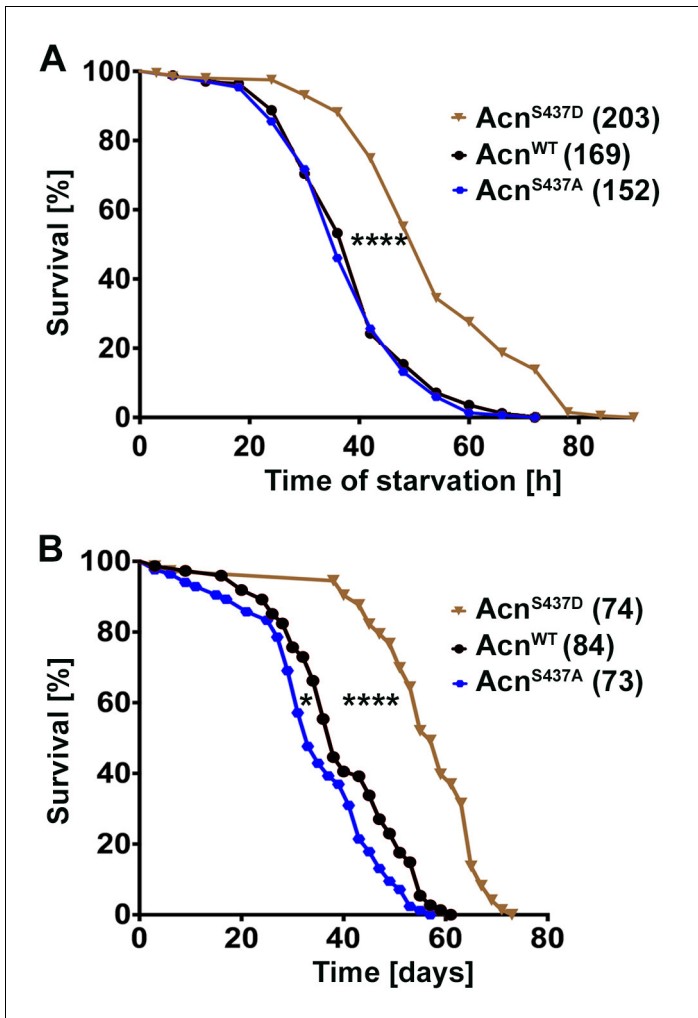

**Figure 3.** Stabilized Acn$^{S437D}$ enhances life span. (A) Starvation-induced mortality of flies expressing the indicated Acn proteins. In parenthesis shown are numbers of initial flies in a single representative experiment out of three. (B) Survival curves of flies expressing the indicated Acn proteins. In parenthesis shown are numbers of initial flies in a single representative experiment out of three. Log-rank comparisons revealed significant differences between survival curves: *$p < 0.05$; ****$p < 0.0001$.
DOI: https://doi.org/10.7554/eLife.30760.012

somewhat faster compared to Acn$^{WT}$ (*Figure 3B*). The median life expectancy for the stabilized Acn$^{S437D}$ mutant was extended by about 50% to 57 days compared to 38 days for Acn$^{WT}$. Together, our data indicate that stabilized phospho-mimetic Acn$^{S437D}$ elevates basal autophagy leading to beneficial outcomes under standard growth conditions.

## Cdk5 phosphorylates Acn at serine 437

To identify the kinase responsible for phosphorylating Acn at serine 437, we performed a targeted RNAi screen. We screened a pre-selected subset of kinases based on hits in software packages [GPS3.0; http://gps.biocuckoo.org (*Xue et al., 2008*) and NetPhosK http://www.cbs.dtu.dk/services/NetPhosK/ (*Miller and Blom, 2009*). To test the ability of these kinases to modify Acn function, we used a sensitized genetic system. Eye-specific GMR-Gal4-driven expression of UAS-Acn$^{WT}$ at 28°C yields a rough-eye phenotype that is susceptible to enhancement or suppression by modifiers of Acn levels (*Nandi et al., 2014*). We reasoned that reduced activity of any kinase responsible for phosphorylating and stabilizing Acn should at least partially suppress the roughness induced by UAS-Acn$^{WT}$. Among the kinases tested, RNAi lines targeting Cdk5 and the MAP kinase p38b exhibited more than 50% suppression of Acn-induced eye roughness (*Figure 4*, *Figure 4—figure supplement 1*, *Supplementary file 1*). Moreover, expression of UAS-Cdk5-RNAi and UAS-p38b RNAi transgenes by themselves did not result in visible phenotypes in the eye (*Figure 4F,M* and *Supplementary file 1*).

We further investigated these two hits from the RNAi screen by examining interactions of gain-of-function and loss-of-function mutants with Acn. GMR-Gal4-driven co-expression of Acn$^{WT}$ with the dominant-negative kinases p38b MAPK$^{K53R}$ or Cdk5$^{K33A}$ effectively suppressed the rough-eye phenotype (*Figure 4C,J*, *Figure 4—figure supplement 2* and *Supplementary file 2*). Furthermore, co-expression of Acn$^{WT}$ with p38b MAPK, Cdk5 or its coactivator p35 (*Tsai et al., 1994*; *Connell-Crowley et al., 2000*) further enhanced eye roughness (*Figure 4D,K,L* and *Supplementary file 2*). By contrast, expression of these indicated Cdk5/p35 and p38b MAPK transgenes by themselves yielded little to no visible eye phenotypes (*Figure 4G,H,N–P* and *Supplementary file 2*). These strong genetic interactions with Acn point to Cdk5 and p38b MAPK as two candidate kinases that may phosphorylate Acn and thereby enhance its activity.

To test whether genetic interactions reflect modification of Serine 437, we used the phospho-specific pS437-Acn antibody to stain eye tissue from wandering third instar larvae. Larvae with Cdk5 or p35 knocked down, as well as *Cdk5* or *p35* mutant larvae exhibited a dramatic reduction of Acn phosphorylation at serine 437 compared to wild-type controls (*Figure 5A–I*). Some brightly pS437-Acn-positive cells remained, however, close to the morphogenetic furrow of Cdk5 or p35 loss-of-function eye discs. In wild-type (*Figure 5B,C*) and Cdk5 loss-of-function eye discs (*Figure 5E,F*), the apical position of these pS437-Acn-positive cells and their shape and DNA distribution identified them as mitotic cells (*Figure 5C,F*). In dividing cells, Acn may be phosphorylated, possibly by mitotically active kinases of the Cdk family with target recognition sequences similar to Cdk5 (*Malumbres and Barbacid, 2009*). Interestingly, Cdk1 knockdown enhanced the Acn overexpression phenotype in eyes (*Supplementary file 1*). As Acn$^{S437A}$ or Acn$^{S437D}$ flies are viable and without obvious mitotic defects, we did not further analyze the significance of elevated pS437-Acn levels in mitotic cells.

Importantly, in *Cdk5* mutant larvae, Acn phosphorylation at serine 437 was restored to wild-type levels with a genomic Cdk5 transgene (*Figure 5J*). Furthermore, overexpression of p35 drastically enhanced Acn-S437 phosphorylation (*Figure 5K*). By contrast, in *p38b* mutant larvae pS437-Acn levels were not altered compared to wild type (*Figure 5A,L*). These data suggest that although both Cdk5 and p38b MAPK genetically interact with Acn, only Cdk5/p35 mediates phosphorylation of Acn-S437.

To further test the in vivo importance of S437 phosphorylation by Cdk5, we compared GMR-Gal4-driven co-expression of UAS-p35 or UAS-p38b MAPK with UAS-Acn$^{WT}$ or UAS-Acn$^{S437A}$, respectively. At 25°C, UAS-Acn$^{WT}$ or UAS-Acn$^{S437A}$ expression yielded mildly rough eyes (*Figure 6A, B*), and expression of only p35 or p38b MAPK did not alter eye morphology (*Figure 6C,D*). Notably, co-expression of p35 enhanced UAS-Acn$^{WT}$-induced roughness significantly more than that of Acn$^{S437A}$ (*Figure 6E,F,I*; p<0.0001; Chi-square test). By contrast, both UAS-Acn$^{WT}$ and UAS-Acn$^{S437A}$ rough-eye phenotypes were enhanced by p38b MAPK co-expression (*Figure 6G,H,I*). Furthermore, Acinus S437 phosphorylation by Cdk5 was also observed in vitro, when purified Acinus proteins,

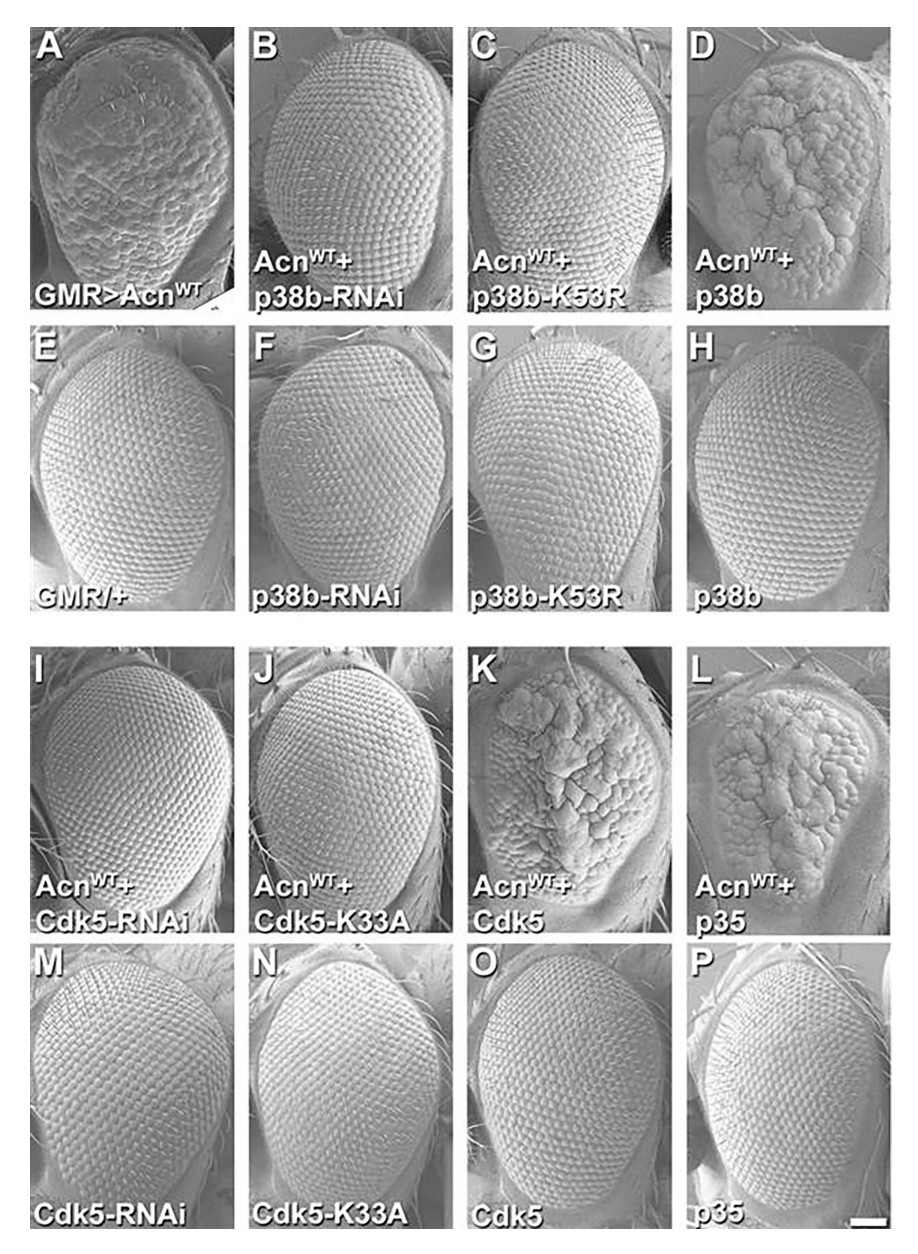

**Figure 4.** Acn genetically interacts with p38b MAP kinase and Cdk5/p35. SEM images of eye expressing the indicated transgenes under GMR-Gal4 control at 28°C. (**A**) Expression of UAS-Acn$^{WT}$ causes a rough eye. This eye-roughness is suppressed by knockdown of p38b MAPK (**B**) or co-expression of dominant-negative p38b MAPK$^{K53R}$ (**C**). By contrast, co-expression of wild-type p38b MAPK enhances the roughness (**D**). Expression of Gal4 (**E**) or the indicated p38b MAPK transgenes in the absence of UAS-Acn$^{WT}$ (**F–H**) causes little or no changes in eye morphology. Eye-roughness induced by UAS-Acn$^{WT}$ is suppressed by knockdown of Cdk5 (**I**) or co-expression of dominant-negative Cdk5$^{K33A}$ (**J**). By contrast, co-expression of wild-type Cdk5 (**K**) or the required cofactor p35 (**L**) enhance Acn$^{WT}$-induced roughness. Expression of the indicated Cdk5 and p35 transgene in the absence of UAS-Acn$^{WT}$ (**M–P**) causes little or no changes in eye morphology. Scale bar in A-P: 50 µm. Quantification of genetic interactions is shown in *Supplementary files 1* and *2*. Detailed genotypes are listed in *Supplementary file 3*.
DOI: https://doi.org/10.7554/eLife.30760.013

The following source data and figure supplements are available for figure 4:

**Source data 1.** Relates to *Figure 4—figure supplement 1*.
DOI: https://doi.org/10.7554/eLife.30760.016

**Figure supplement 1.** Knockdown efficiency of p38b MAPK, Cdk5 and p35 RNAi UAS-transgenes.

*Figure 4 continued on next page*

*Figure 4 continued*

DOI: https://doi.org/10.7554/eLife.30760.014

**Figure supplement 2.** Expression of dominant-negative mutant Cdk5 and p38b MAP kinases.

DOI: https://doi.org/10.7554/eLife.30760.015

either bacterially-expressed GST-Acn$^{402-527}$ fusion proteins (*Figure 6J*) or S2 cell-expressed full-length Acinus proteins (*Figure 6K*) were exposed to purified Cdk5/p35 kinase complex. Together these findings indicate that, unlike for p38b MAPK, Acn S437 is the physiologically relevant residue targeted by the Cdk5/p35 complex.

## Acn regulates basal autophagy and life span by Cdk5/p35-dependent phosphorylation

Flies mutant for the Cdk5 co-activator *p35* display adult onset neurodegeneration and reduced life-span (*Connell-Crowley et al., 2007*; *Trunova and Giniger, 2012*). To test whether altered basal autophagy contributes to these phenotypes, we first examined the distribution of Atg8a in eye discs from fed 96-hr old larvae. Compared to wild type (*Figure 7A*), *p35* mutant eye discs displayed a significantly reduced number of Atg8a-positive punctae (*Figure 7B,K*). Basal autophagy in *p35* eye discs was restored to wild-type level by expression of phospho-mimetic Acn$^{S437D}$, but not Acn$^{WT}$ or Acn$^{S437A}$, under control of the endogenous *acn* promoter (*Figure 7C–E,K*). Cellular levels of the autophagy receptor p62 depend on autophagic flux and thus are elevated in cells with impaired basal autophagy (*Pircs et al., 2012*; *DeVorkin and Gorski, 2014b*). By immunostaining, p62 levels were significantly elevated in *p35* eye discs compared to wild type (*Figure 7F,G,L*), consistent with impaired basal autophagy. Accumulation of p62 in *p35* eye discs was not prevented by expression

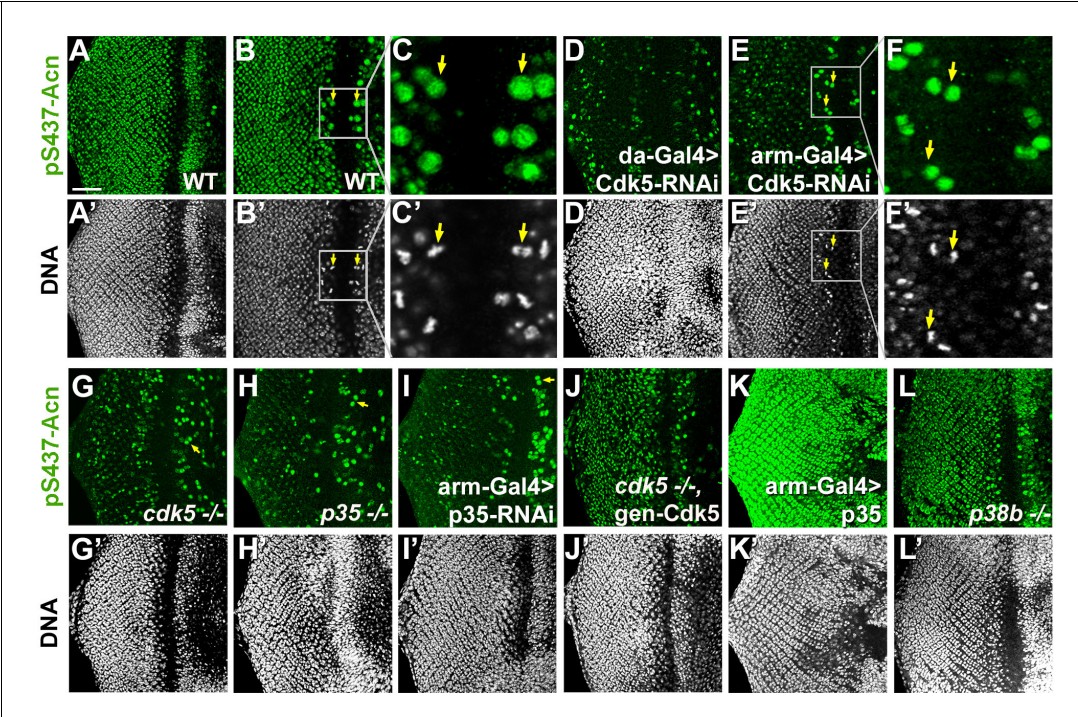

**Figure 5.** Cdk5/p35-mediates phosphorylation of Acinus. Projections (**A, D, G–L**) or individual optical sections (**B,C,D,E**) of confocal micrographs of eye discs stained for pS437-Acn or DNA from wild-type controls (**A–C**), or from larvae with knockdown for Cdk5 (**D–F**), mutant for *Cdk5* (**G**) or *p35* (**H**), or with p35 knockdown (**I**), or *Cdk5* mutant larvae rescued with a genomic Cdk5 transgene (**J**), from larvae overexpressing p35 (**K**) or mutant for p38b MAP kinase (**L**). Arrows indicate examples of mitotic cells with high pS437-Acn levels. These are best seen in individual optical sections of wild-type (**B,C**) or Cdk5-RNAi eye discs (**E,F**). Scale bar is 50 µm in A, D,E, G-L; 32 µm in B; 10 µm in C; and 15 µm in F.

DOI: https://doi.org/10.7554/eLife.30760.017

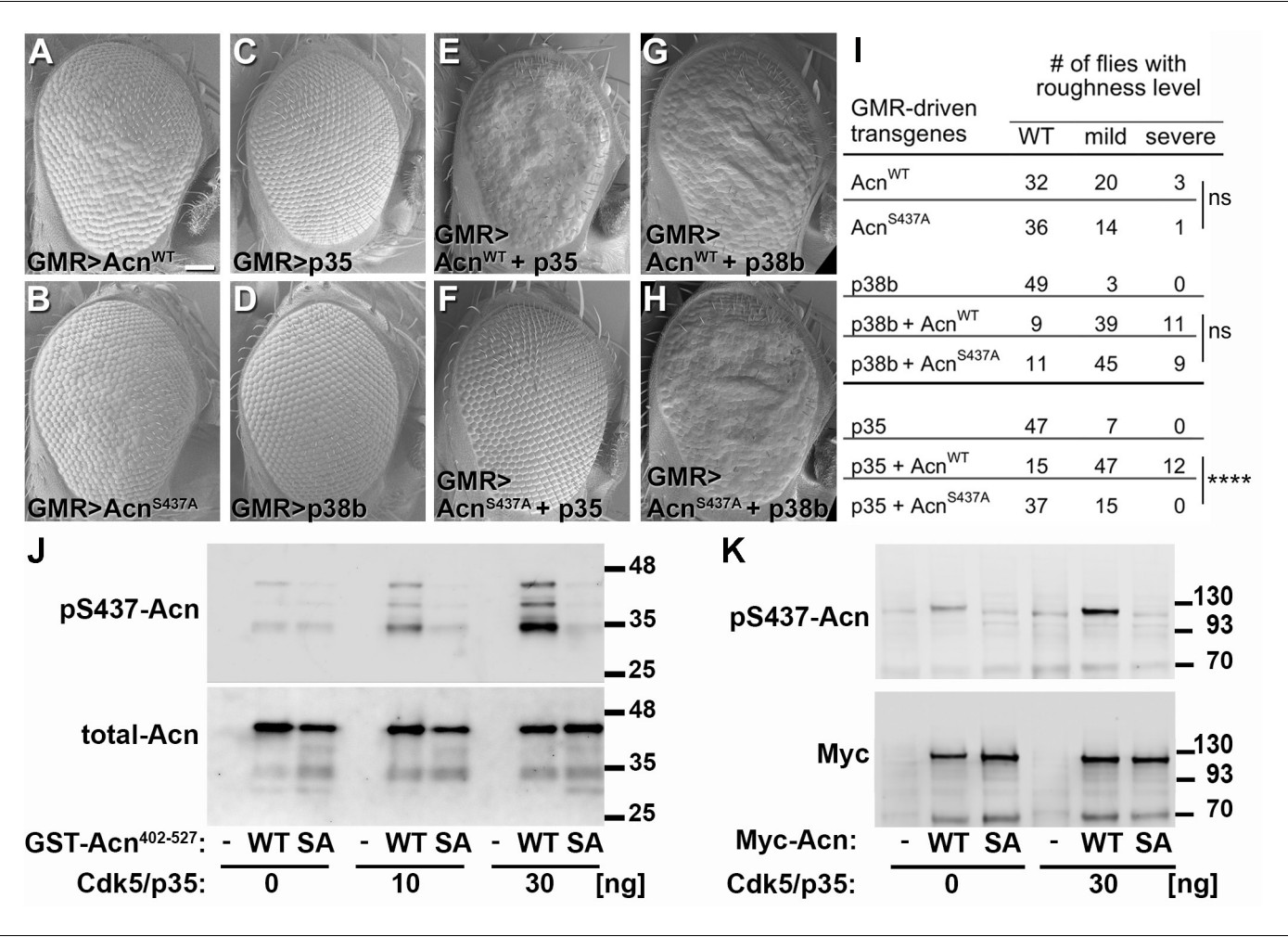

**Figure 6.** Acn-S437 is the critical target site for Cdk5/p35-mediated phosphorylation. (A–H) SEM images of eyes expressing the indicated transgenes under UAS/GMR-Gal4 control at 25°. Under these conditions, expression of Acn^WT (A) or Acn^S437A (B) causes a mildly rough eye, but not expression of p35 (C) or p38b MAP kinase (D). Coexpression of p35 with Acn^WT (E) causes a severely rough eye, but not p35 coexpression with Acn^S437A (F). By contrast, p38b MAP kinase coexpression enhances roughness of both, Acn^WT (G) and Acn^S437A (H). Quantification of these genetic interactions is shown in panel I. Statistical significance was calculated by Chi square test (****p<0.0001; ns, not significant). (J–K) Western blots of purified wild-type (WT) or S437A mutant (SA) GST-Acn^402-527 fusion proteins (J) or full-length Streptag-Myc-Acn proteins (K) that were incubated with the indicated amount of Cdk5/p35 kinase complex. Blots were developed with antibodies against pS437-Acn (1:2000), Myc (1:2000), or total Acn (1:2000) as described (*Nandi et al., 2014*). Note that the Acn antibody (*Haberman et al., 2010*) preferentially recognizes a C-terminal Acn epitope that is deleted in some of the partially degraded GST-fusion proteins, which however still contain the Cdk5 target site at S437. Detailed genotypes are listed in *Supplementary file 3*.

DOI: https://doi.org/10.7554/eLife.30760.018

of Acn^WT or Acn^S437A (*Figure 7H,I,L*), whereas Acn^S437D expression restored wild-type p62 levels (*Figure 7J,L*).

Next, we explored the physiological consequences of manipulating levels of basal autophagy in *p35* mutants. Consistent with previous reports (*Connell-Crowley et al., 2007*), *p35* mutants had reduced life expectancy compared to wild type (*Figure 7M*). Near wild-type life span was restored in *p35* mutants that express phospho-mimetic Acn^S437D, but not Acn^S437A or Acn^WT (*Figure 7M*). Together, these data indicate that Acn-S437 is a physiologically relevant target for Cdk5/p35-mediated phosphorylation.

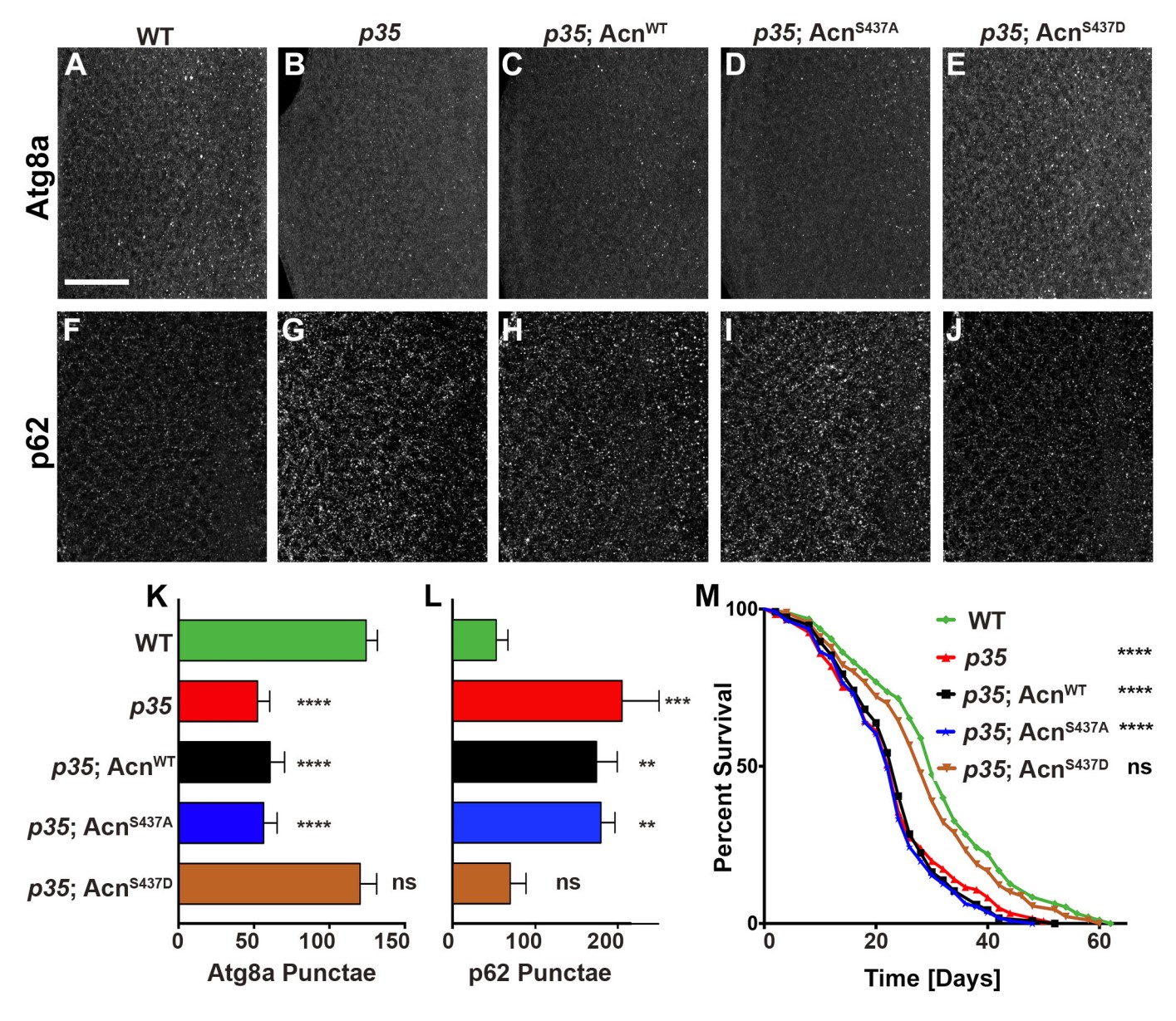

**Figure 7.** Cdk5/p35-mediated phosphorylation of Acn regulates basal autophagy and longevity. Projections of confocal micrographs of eye discs stained for Atg8a (A–E) or p62 (F–J) from wild-type controls (A,F), *p35* mutant larvae (B,G) or *p35* mutant larvae expressing Acn[WT] (C,H), Acn[S437A] (D,I) or the phosphomimetic Acn[S437D] (E,J) under control of the *acn*P promoter. Scale bar in A is 40 µm in A-J. (K) Quantification of Atg8a punctae from three eye discs each with genotypes as shown in A-E. (L) Quantification of p62 punctae from three eye discs each with genotypes as shown in F-J. Bar graphs show mean ±SD. ns, not significant; **p<0.01; ***p<0.001; ***p<0.0001; each compared to wild-type Acn control. (M) Survival curves of WT controls or *p35* mutants, or *p35* mutants expressing the indicated Acn proteins. The initial numbers of flies were 95 (WT), 121 (p35), 116 (*p35*, Acn[WT]), 111 (*p35*, Acn[S437A]), 90 (*p35*, Acn[S437D]). Log rank tests compared to WT control: ns, not significant; ****p<0.0001. Detailed genotypes are listed in *Supplementary file 3*.

DOI: https://doi.org/10.7554/eLife.30760.019

The following source data is available for figure 7:

**Source data 1.** Quantification of Atg8a and p62 punctae in eye discs.
DOI: https://doi.org/10.7554/eLife.30760.020

# Protein aggregation triggers phosphorylation of Acn S437

To maintain normal life span, autophagy is required in neurons for the suppression of neurodegeneration. To test a possible role of Cdk5-mediated Acn phosphorylation in this context, we used two different *Drosophila* Huntington's disease models expressing huntingtin-polyQ polypeptides in the eye, either through GMR-Gal4-driven expression of UAS-Htt.Q93 (*Figure 8B,C*) or through a direct fusion of Htt-Q120 with the GMR enhancer/promoter region (GMR-Htt.Q120, *Figure 8D,E*). Compared to wild-type eye discs (*Figure 8A*), Acn-S437 phosphorylation was elevated in posterior cells

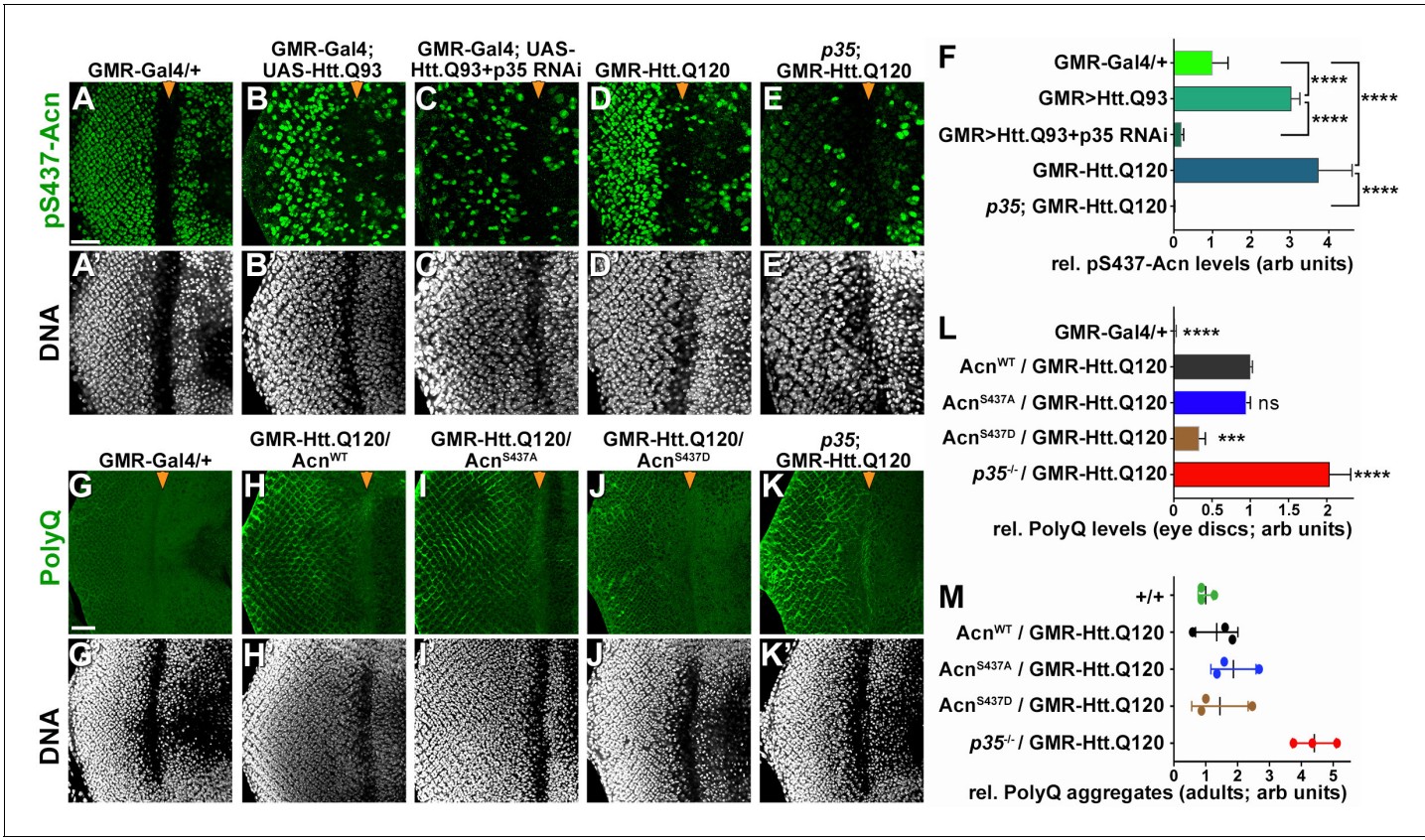

**Figure 8.** Protein-aggregate-induced phosphorylation of Acn S437 depends on Cdk5/p35. Projections of confocal micrographs of eye discs stained for pS437-Acn (A–E) or polyQ proteins (G–K) and DNA. Compared to a GMR-Gal4 control (A), GMR-Gal4-driven UAS-Htt.Q93 (B) or GMR-Htt.Q120 (D) expression induced elevated S437 phosphorylation, which was suppressed in eye discs with p35 knockdown (C) or mutant for *p35* (E). GMR-directed expression initiates at the furrow (arrowhead) and is more developed toward the posterior. (F) Quantification of S437 phosphorylation (F) averaged constant areas containing about 50 ommatidial clusters located at least 6–8 rows posterior to the furrow. Bar graphs show mean ±SD of integrated densities. Values were normalized to GMR-Gal4 controls and were from one representative experiment out of three repeats. ns, not significant; **p<0.01; ***p<0.001; ***p<0.0001; for indicated comparisons. Early polyQ accumulation was compared between GMR-Gal4 eye discs as controls (G), and eye discs expressing GMR-Htt.Q120 and carrying a copy of the indicated genomic Acn transgene (H–J) or the p35 null allele (K). Compared to Acn$^{WT}$ (H) and Acn$^{S437A}$ (I) eye discs, in Acn$^{S437D}$ (J) polyQ accumulation was reduced until some 7 to 8 rows of ommatidia posterior to the furrow (arrowhead) and enhanced in p35 mutants (H). Scale bar in A: 40 μm in A-N. Posterior is to the left. (L) Quantification of PolyQ accumulation averaged constant areas containing about 100 ommatidial clusters located at least 2–3 rows posterior to the furrow. Bar graphs show mean ± SD of integrated densities. Values were normalized to Acn$^{WT}$/GMR-Htt.Q120 (L) and were from one representative experiment out of three repeats. ns, not significant; **p<0.01; ***p<0.001; ***p<0.0001; compared to Acn$^{WT}$/GMR-Htt.Q120. (M) Quantification of dot blots measuring aggregated polyQ protein in fly heads expressing GMR-Htt.Q120 and the indicated genomic Acn transgenes or in a *p35* mutant background. Control OreR flies (+/+) indicate background level of dot blot measurements (n = 3 biological repeats). The scatter plot shows mean and standard deviation from three separate experiments. Detailed genotypes are listed in *Supplementary file 3*.

DOI: https://doi.org/10.7554/eLife.30760.021

The following source data is available for figure 8:

**Source data 1.** Quantification of pS437 and polyQ-levels in eye discs.
DOI: https://doi.org/10.7554/eLife.30760.022

of Htt-polyQ expressing eye discs (*Figure 8B,D,F*). This activation of Acn was dependent on Cdk5/p35 as it was suppressed in eye discs with p35 knocked down (*Figure 8C,F*) or mutant for *p35* (*Figure 8E,F*).

To examine the consequence of Acn activation for polyQ accumulation, we stained eye discs for polyQ proteins. GMR-Htt.Q120 expression resulted in its accumulation a few rows posterior to the furrow in eye discs expressing Acn^WT or Acn^S437A under control of the *acn* promoter (*Figure 8G–I,L*). By contrast, expression of the stabilized phospho-mimetic Acn^S437D protein yielded an initial reduction in polyQ accumulation just posterior to the furrow (*Figure 8J,L*). This is consistent with the data above that show elevated autophagy in Acn^S437D eye discs (*Figure 2A,C*) and with the known role of autophagy in the clearance of protein aggregates (*Menzies et al., 2017*). In *p35* mutant eye discs, polyQ protein levels were further elevated (*Figure 8K,L*), which resulted in the accumulation polyQ aggregates in adults as detected by a filter retardation assay (*Figure 8M*).

Acn activation was not specific for Htt-polyQ peptides, as overexpression of other neurodegeneration linked polyQ proteins, specifically SCA3.Q78 (*Figure 9B,G*) and ATX1.Q82 (*Figure 9C,G*) yielded similar increases in Acn-S437 phosphorylation. Furthermore, expression of human Aβ42 peptide resulted in elevated levels of S437 phosphorylated Acn (*Figure 9D,G*). By contrast, overexpression of Parkinson's disease-linked alpha-Synuclein (*Figure 9E,G*) or Amyotrophic Lateral Sclerosis (ALS)-linked human SOD1 (*Figure 9F,G*) in larval eye disc failed to elevate Acn S437 phosphorylation compared to wild type (*Figure 9A*).

## Discussion

Numerous studies have implicated dysregulation of Cdk5 activity in neurodegenerative diseases due to its role in regulating cytoarchitecture, axonal transport, and synaptic activity (*Su and Tsai, 2011*; *Cheung and Ip, 2012*; *Shah and Lahiri, 2017*). Here, we show that reduced Cdk5 activity can also reduce neuronal fitness by compromising basal autophagy. We find that the effect of the Cdk5/p35 complex on autophagy depends on its role in phosphorylating the conserved S437 in Acn. Interfering with Acn-S437 phosphorylation, either by loss of Cdk5/p35 function, or by mutation of its target site in Acinus, reduces the level of basal, starvation-independent autophagy and shortens life span. Importantly, these phenotypes were reversed by the phospho-mimetic Acn^S437D mutation. The beneficial outcomes that result from Acinus stabilization, including the extended lifespan and suppression of polyQ protein accumulation (our data and *Nandi et al., 2014*), argue that increases in autophagic flux and autolysosome size in stabilized phospho-mimetic Acn^S437D mutants are not a response to a proteotoxic stress or reflective of a defect in lysosomal function due to Acn^S437D expression, but reflect a beneficial activation of autophagy as previously observed in multiple systems (e.g. *Sarkar et al., 2007*; *Simonsen et al., 2008*; *Menzies et al., 2015*; *Gelino et al., 2016*). Therefore, these findings indicate the importance of Cdk5-mediated Acn-S437 phosphorylation for maintaining neuronal health.

How does phosphorylation of S437 stabilize Acinus and boost its function? One possibility is a reduction in caspase-mediated cleavage of Acinus (*Sahara et al., 1999*). Inhibition of this cleavage has previously been shown as a consequence of Akt1-mediated Acinus phosphorylation in apoptotic (*Hu et al., 2005*) and also in non-apoptotic cells (*Nandi et al., 2014*) and furthermore, upon binding of the anti-apoptotic protein AAC11 (*Rigou et al., 2009*). Our data, however, argue against this possibility. We have previously shown that preventing its caspase-mediated cleavage in the Acn^D527A mutant stabilized Acn only in a subset of photoreceptor cells (predominantly R3 and R4, *Nandi et al., 2014*). By contrast, S437 phosphorylation elevates Acinus levels in the majority of photoreceptor cells (*Figure 1*). We thus favor the notion of a different Acn cleavage, closer to the S437 residue. Such a cleavage has been reported for mammalian Acinus (*Sahara et al., 1999*) at a residue corresponding to A423 in Drosophila Acn. It will be important to identify the protease responsible for this cleavage and test its impact on Acn function.

In the context of neurodegenerative diseases, much interest has been focused on the aberrant activation of Cdk5. Elevated Cdk5 activity can be induced by multiple stressors such as inflammation, ischemia, or mitochondrial dysfunction (*Su and Tsai, 2011*). These stressors can trigger pathological activation of Cdk5 by calpain-mediated cleavage of its p35 or p39 co-activators (*Lee et al., 2000*). The cleavage product p25 lacks the myristoylation-tag that anchors activated Cdk5 in complexes with p35 or p39 to membranes and escapes the inhibitory auto-phosphorylation of p35/39 that limits

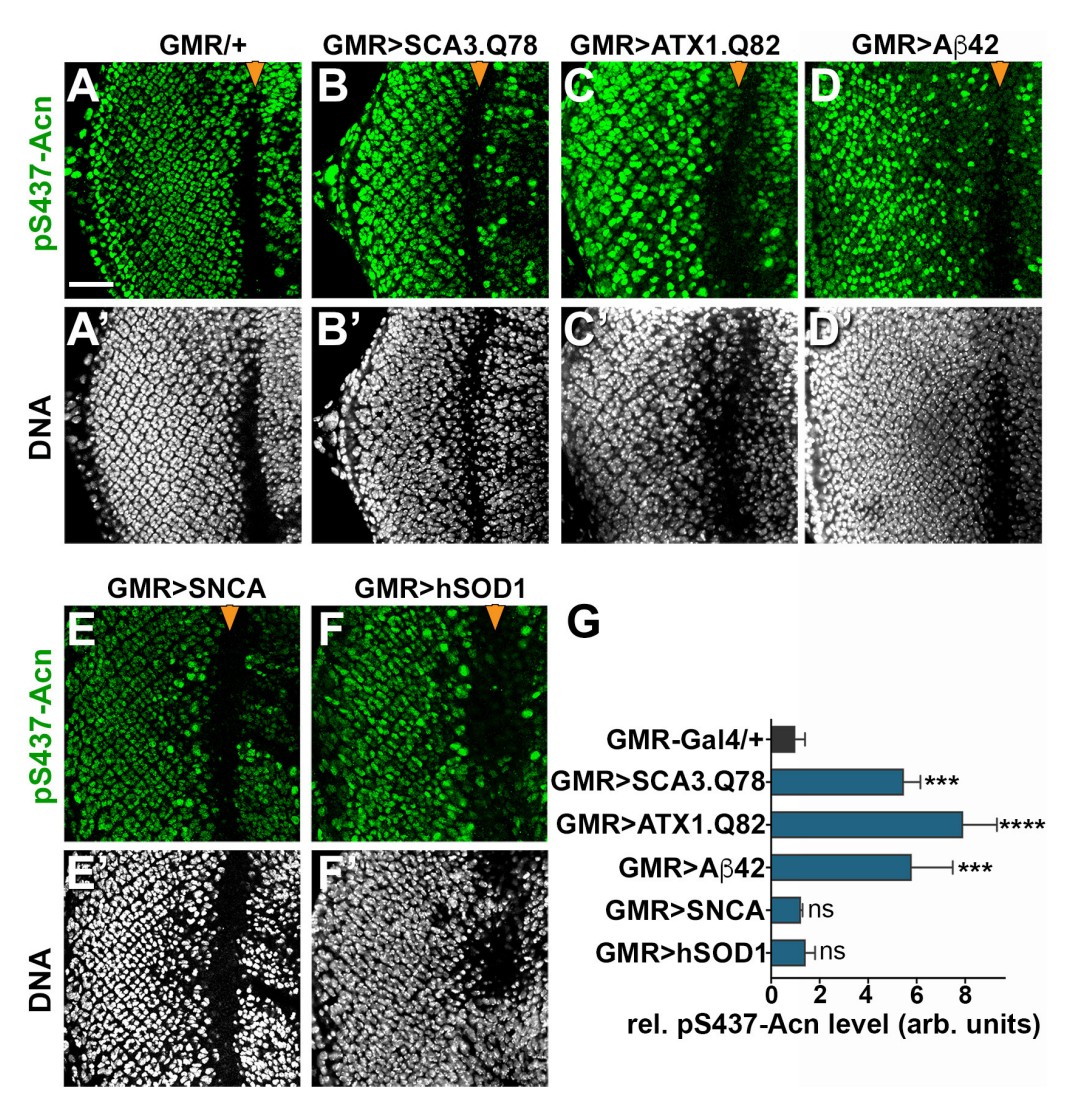

**Figure 9.** Increased Acn-S437 phosphorylation is specific for a subset of neurodegeneration models. Projections of confocal micrographs of eye discs stained for pS437-Acn (A–F) or DNA (A'–F'). Compared to GMR-Gal4 controls (A), eye discs with GMR-Gal4-driven UAS-SCA3.Q78 (B) or UAS-ATX1.Q82 (C) or UAS-Aβ42 (D) display elevated levels of pS437-Acn staining in a subset of cells, in contrast to UAS-alpha-Synuclein (E) or UAS-hSOD1 (F) which appear unaltered compared to controls. GMR-directed expression of transgenes initiates at the furrow (arrowhead) and extends toward the posterior (left). Scale bar in A is 40 μm in A-N. Detailed genotypes are listed in *Supplementary file 3*. (G) Quantifications of S437 phosphorylation averaged constant areas containing about 50 ommatidial clusters located at least 6–8 rows posterior to the furrow. Bar graphs show mean ±SD of integrated densities of pS437 staining normalized to GMR-Gal4 controls. ns, not significant; **p<0.01; ***p<0.001; ***p<0.0001; compared to GMR-Gal4/+.

DOI: https://doi.org/10.7554/eLife.30760.023

The following source data is available for figure 9:

**Source data 1.** Quantification of pS437-Acn in neurodegenerative model eye discs.
DOI: https://doi.org/10.7554/eLife.30760.024

their life time in active complexes (*Patrick et al., 1999*). The resulting unrestrained phosphorylation of proteins involved in microtubule-based axonal transport and synaptic proteins contributes to the progression of ALS, Alzheimer's diseases and other neurological diseases (*McLinden et al., 2012*; *Klinman and Holzbaur, 2015*).

In *Drosophila*, we observe Cdk5-depedent elevated Acn-S437 phosphorylation in response to the expression of polyQ proteins, but we do not know yet whether this increase reflects elevated Cdk5/p35 activity or pathological Cdk5/p25 activity following calpain-mediated cleavage of p35. Alternatively, the phosphatases that remove the phosphate group from pS437- Acn may be inhibited. Acn-S437 phosphorylation is highly dynamic in developing eyes (*Figure 1*), but phosphatases acting on pS437-Acn have not yet been identified. Interestingly, in addition to multiple polyQ proteins, Aβ42 expression is another *Drosophila* neurodegenerative disease model that triggered elevated phosphorylation of Acn-S437. This is consistent with increased Cdk5 activity reported for multiple Alzheimer's disease models (*Otth et al., 2002*; *Shah and Lahiri, 2017*) which in turn may contribute to the Cdk5-dependent phosphorylation of Tau as a possible contribution to the progression of Alzheimer's disease (*Cruz et al., 2003*).

Not all neurodegeneration models induced Cdk5 activity as visualized by Acn-S437 phosphorylation. Pathological Cdk5 activation is believed to contribute to Parkinson's disease (*Smith et al., 2006*). Nevertheless, overexpression of alpha-Synuclein, which mimics some aspects of Parkinson's disease in *Drosophila* (*Kontopoulos et al., 2006*), is not sufficient to increase pS437-Acn levels. Similarly, Acn phosphorylation was unchanged in response to hSOD1 over-expression in a *Drosophila* model of ALS (*Watson et al., 2008*). Both of these models trigger substantial cellular stress and rapid neurodegeneration (*Jaiswal et al., 2012*), indicating that Acn phosphorylation is not a generic response to cellular stress, but more likely involves specific activation of Cdk5.

An example of such a specific interaction was recently elucidated in the context of polyQ proteins. *Ashkenazi et al. (2017)* showed that the levels of the autophagy regulator Beclin1 are maintained by its Ataxin3-mediated de-ubiquitination. Their interaction is mediated by the polyQ tract in wild-type Ataxin3 and inhibited by the presence of pathological polyQ tracts known to trigger Huntington's disease or SCA3 (*Ashkenazi et al., 2017*). Furthermore, an increasing number of autophagy receptors are being identified that drive selective autophagy of specific cargoes (*Rogov et al., 2014*; *Khaminets et al., 2016*). Interestingly, several of these receptors are activated by modifications such as phosphorylation or ubiquitination in response to specific stressors, such as polyQ proteins (*Deng et al., 2017*). To which extend such modifications of autophagy receptors contribute to the induction of basal autophagy by Acn remains to be tested.

Intriguingly, Cdk5-mediated phosphorylation stabilizes Huntingtin (*Luo et al., 2005*) and thereby may promote its function as a scaffold for selective autophagy (*Ochaba et al., 2014*; *Rui et al., 2015*). Cdk5 may thus act through different effectors in different diseases. Endophilin B1 was identified as a Cdk5 substrate, the phosphorylation of which is required for starvation-induced autophagy (*Wong et al., 2011*). Importantly, Cdk5-mediated phosphorylation of Endophilin B1 appeared necessary for the elimination of dopaminergic neurons in an MPTP mouse model of Parkinson's disease (*Wong et al., 2011*). Furthermore, MEKK1 is a key target for Cdk5 in a *Drosophila* model of retinitis pigmentosa (*Kang et al., 2012*). Such diversity of targets may not only apply to different diseases, but also to different stages of a given disease.

Although the pathological activation of Cdk5 activity appears to be a contributing factor to the progression of some neurodegenerative diseases, we show that wild-type levels of Cdk5/p35 activity are necessary to support basal autophagy in the clearance of protein aggregates. We show that this effect, at least in part, is due to Cdk5-mediated phosphorylation of Acn as phospho-mimetic Acn$^{S437D}$ reverses the reduced basal autophagy and the shortened life span observed in *p35* mutants due their adult-onset, progressive neurodegeneration (*Connell-Crowley et al., 2007*; *Trunova and Giniger, 2012*).

How does the Cdk5-mediated phosphorylation of Acn-S437 elevate the level of basal autophagy? We previously have shown that levels of Acn are critical for regulating basal autophagy in an Atg1-dependent manner (*Nandi et al., 2014*). High Acn levels can drive excessive autophagy, even in the presence of activated mTor, and cause autophagy-dependent lethality (*Haberman et al., 2010*). More subtle effects resulted when inhibiting caspase-mediated cleavage or phospho-mimetic mutations elevated levels of Acinus expressed from its own promoter (*Nandi et al., 2014*, *Figure 1*). For example, the phosphorylation of two conserved Akt1-target sites has been shown to elevate Acn levels (*Hu et al., 2005*; *Nandi et al., 2014*). In *Drosophila*, this stabilized Acn promotes starvation-independent basal autophagy (*Nandi et al., 2014*). Similarly, we find that Cdk5-mediated phosphorylation stabilizes Acn protein and promotes autophagy (*Figure 7*), as does the phospho-mimetic Acn$^{S437D}$ mutation (*Figure 2*). Nevertheless, stabilized Acinus proteins do not reach the levels

necessary to cause the developmental defects that can be triggered by overexpression through the Gal4/UAS system (*Haberman et al., 2010*; *Nandi et al., 2014*).

Nuclear localization of Acn as a Cdk5 substrate seems to be in conflict with the localization of the activated Cdk5/p35 complex to the plasma membrane due the myristoylation of p35 (*Patrick et al., 1999*). However, non-myristoylated p35 and p39 preferentially accumulate in the nucleus and can bind and activate nuclear Cdk5 (*Asada et al., 2008*). How nuclear or possibly cytoplasmic Acn induces autophagy remains unclear. Phosphorylated and unphosphorylated Acn proteins are primarily nuclear and neither phosphorylation at S437 nor the two Akt1 target sites S641 and S731 are necessary for starvation-induced autophagy, although they enhance basal autophagy (*Figure 2* and *Nandi et al., 2014*). Acn is a required component of the nuclear ASAP complex (*Schwerk et al., 2003*; *Murachelli et al., 2012*) which participates in the regulation of alternative splicing (*Hayashi et al., 2014*; *Malone et al., 2014*). Future work will therefore focus on attempts to identify specific Acn-dependent transcripts that may play a role in the regulation of autophagy or identifying alternative mechanisms for its role in the regulation of basal, starvation-independent autophagy.

## Materials and methods

A key resources table can be found in *Supplementary file 4*.

### Fly work

Flies were maintained using standard conditions. Bloomington Stock Center provided Da-Gal4, Arm-Gal4, GMR-Gal4 driver lines, $w^{1118}$, the RNAi lines and human neurodegenerative disease model lines (BS lines: 33769, 8141, 33818, 51376, 33606, 8534). Other fly strains used were $p38b^{\Delta45}$, a null generated by transposon excision and removing most of the p38b coding region, UAS-p38b, UAS-p38b$^{K53R}$ (*Vrailas-Mortimer et al., 2011*); $p35^{20C}$, which deletes ~90% of the *p35* coding region, including all sequences required for binding to and activating Cdk5; (*Connell-Crowley et al., 2007*), UAS-p35, UAS-Cdk5, UAS-Cdk5$^{K33A}$; (*Connell-Crowley et al., 2000*). A *Cdk5* null allele and genomic rescue transgene (*Kissler et al., 2009*) were gifts from Edward Giniger, National Institute of Neurological Disorders and Stroke, Bethesda, Maryland. UAS-Htt-exon1-Q93, (*Steffan et al., 2001*), abbreviated UAS-Htt.Q93, was a gift from Robin Hiesinger, Free University Berlin, Berlin, Germany. Transgenic flies were generated by BestGene, Inc. DNA constructs related to genomic *acn* were generated by standard mutagenesis of a 4 kb Acn DNA fragment sufficient for genomic rescue (*Haberman et al., 2010*), confirmed by sequencing, cloned into an Attb vector, and inserted into the 96F3 AttP landing site (*Venken et al., 2006*). Similarly, UAS-controlled wild-type and mutant Acn transgenes were generated by standard mutagenesis from full-length Acn cDNA, confirmed by sequencing, and inserted into pUAS vectors modified by addition of an AttB site (*Nandi et al., 2014*). Experiments with UAS-RNAi transgenes were performed at 28°C to maximize knockdown efficiency.

Starvation resistance and life span were analyzed as described previously (*Nandi et al., 2014*). Briefly, for starvation resistance 4- to 5-day-old virgins were kept in vials containing 1% agarose in 1X PBS at 25°C and dead flies are counted every 6 hr intervals. To measure life spans, males that emerged within a 2-day period were aged for an additional 3 days, kept in demographic cages and their survival at 25°C was recorded every other day.

### Biochemistry

Antibodies against pS437-Acn was raised in rabbits by Genemed Synthesis against the Acn peptide H I V R D P- S(p)-P A R N R A S and double-affinity purified. For Western blots, five 96 hr larvae were crushed in 300 µl lysis buffer (10% SDS, 6 M urea, and 50 mM Tris-HCl, pH 6.8) at 95°C, boiled for 2 min, and spun for 10 min at 20,000x*g*. 20 µl lysate from larvae were separated by SDS-PAGE, transferred to nitrocellulose membranes, blocked in 3% non-fat dried milk and probed with mouse antibodies against Actin (JLA20) or Myc (9E10; both at 1:2,000; Developmental Studies Hybridoma Bank), guinea pig anti-Acn (aa 423–599, 1:3,000, *Haberman et al., 2010*). For Western blots with pS437-Acn antibodies, 1% BSA was used as blocking reagent. Using IR-dye labeled secondary antibodies and the Odyssey scanner (LI-COR Biosciences) bound antibodies were detected and quantified by comparison to Actin. Prestained molecular weight markers (HX Stable) were obtained from UBP-Bio.

Non-radioactive Cdk5 in vitro kinase assays were performed essentially as described (*Hong and Guan, 2017*). In brief, GST-Acn$^{402-527}$ fusion proteins (WT or S437A mutant) were expressed in a 30-ml culture of BL21 bacteria and immobilized and purified on 10 µl glutathione sepharose beads using standard procedures (*Hong and Guan, 2017*). Alternatively, Streptag-2xMyc-tagged full-length Acinus proteins (WT or S437A) were expressed in S2 cells as described (*Stenesen et al., 2015*), immobilized and purified on Strep-Tactin magnetic beads (IBA) according to the manufacturer's instructions.

Immobilized GST-Acn fusion proteins or Streptag-Myc-tagged Acn proteins were exposed to 10 or 30 ng Cdk5/p35 complex (Upstate (Sigma) 14–477) in 30 µl kinase assay buffer (25 mM MOPS, pH 7.2, 12. 5 mM glycerol 2-phos-phate, 25 mM MgCl$_2$, 5 mM EGTA, 2 mM EDTA, 0.25 mM DTT, and 0.5 mM ATP) for 20 min at 30℃. Immobilized Acn proteins were washed twice with PBS containing 0.1% Triton-X100, eluted with 20 mM glutathione and analyzed by western blots using antibodies against Acinus (1:2000, *Haberman et al., 2010*) or pS437-Acn (1:2000).

Quantitative RT-PCR was used to measure transcript levels of Myc-tagged Acn transgenes and knockdown efficiencies as previously described (*Akbar et al., 2011*). In short, RNA was isolated using TRIZOL (Ambion) according to the manufacturer's instructions. 2 µg RNA was reverse transcribed using High-Capacity cDNA Reverse Transcription kit (Applied Biosystems) using random hexamer primers. Quantitative PCR was performed using the Fast SYBR Green Master Mix in a real-time PCR system (Fast 7500; Applied Biosystems). Each data point was repeated three times and normalized for the message for ribosomal protein 49 (RP49).

Primers were:
Myc-left, 5′-CTGGAGGAGCAGAAGCTGAT-3′, within the Myc and
Acn_right, 5′-GGAGTCTCGACCTCGGTCTT-3′, within the Acn coding regions, and
RP49_left, 5′-ATCGGTTACGGATCGAACAA-3′, and
RP49_right, 5′-GACAATCTCCTTGCGCTTCT-3′.
Cdk5_left: 5′-AATGGAGAAGATCGGGGAGG-3′
Cdk5_right: 5′-GGGAGATCTGCC TGCTGA-3′
p38b_left: 5′-GACGCCGATCTGAACAACAT-3′
p38b_right: 5′-ATCCTGGATTTCGGTTTGGC-3′
p35_left: 5′-TGTTCTTGCACTGTCGTTGT-3′
p35_right: 5′-TCAGCGGAGAAGAGAGCAAG-3′

## Histology

SEMs of fly eyes were obtained as previously described (*Wolff, 2011*). Briefly, eyes were fixed in 2% paraformaldehyde, 2% glutaraldehyde, 0.2% Tween 20, and 0.1 M cacodylate buffer, pH 7.4, for 2 hr. Fixed samples were washed for 12 hr each in a series of four washes with increasing ethanol (25–100%). This is followed by a series of hexamethyldisilazane washes (25–100% in ethanol) for 1 hr each. Flies air dried overnight were mounted on SEM stubs and coated in fast-drying silver paint on their bodies only. Flies were sputter coated with a gold/pallidum mixture for 90 s and imaged at 1000 $\times$ magnification, with extra high tension set at 3.0 kV on a scanning electron microscope (SIGMA; Carl Zeiss). The microscope was equipped with the InLens detector (Carl Zeiss).

For TEM, size-matched 96 hr fed larvae were dissected and processed as described earlier in *Nandi et al. (2014)*. In short, dissected larvae were fixed in 2% glutaraldehyde in 0.1 M cacodylate buffer, pH 7.2 and postfixed with 2% OsO$_4$ and 1.5% KFeCN in the same buffer. Samples were embedded in epoxy resin, sectioned. Sections were stained with uranyl acetate and lead citrate to enhance contrast, examined with a transmission electron microscope (120 kV; Tecnai G2 Spirit Bio-TWIN; FEI), and images were captured with an 11-megapixel camera (Morada; Olympus). From TEMs, measurements of autolysosomal diameters and areas were obtained using Macnification software (Orbicule).

## Immunofluorescence

Whole-mount tissues were prepared for immunofluorescence staining as previously described (*Akbar et al., 2011*). Briefly, dissected samples were fixed in periodate-lysine-paraformaldehyde, washed in PBS, permeabilized with 0.3% saponin in PBS (PBSS), blocked with 5% goat serum in PBSS, and stained with the indicated primary antibodies: guinea pig anti-Acn (1:1000,

*Haberman et al., 2010*), mouse anti-Myc (9E10, 1:1000), mouse anti-FLAG (1:1000; Sigma), rabbit anti-p62 (1:2000, *Pircs et al., 2012*), a gift from G. Juhàsz (Eötvös Loránd University, Budapest, Hungary), rabbit anti-GABARAP (1:200; Abcam, ab109364), which detects endogenous Atg8a (*Kim et al., 2015*), mouse-anti 1C2 (1:1,000; MAB1574; EMD Millipore), rabbit anti-Boss (1:2000, *Krämer et al., 1991*), and secondary antibodies were labeled with Alexa Fluor 488, 568, or 647 (1:500; Molecular Probes) and mounted in Vectashield containing DAPI (Vector Laboratories). Fluorescence images were captured with 63×, NA 1.4 or 40×, NA1.3 Plan Apochromat lenses on an inverted confocal microscope (LSM 510 Meta or LSM 710; Carl Zeiss Jena). Confocal Z-stacks of eye discs were obtained at 1 µm step size.

For phosphatase treatment, dissected third instar larval carcasses were fixed in periodate-lysine-paraformaldehyde and treated with 130 U/ml Calf Intestinal Phosphatase (New England Biolabs, Inc.) for 3 hr at 37°C with 1X protease inhibitor tablet (Roche) dissolved in 1X PBS, pH 7.5. Subsequently, tissues were processed and stained, and eye disc mounted as described above. For autophagy flux experiments, 72-hr-old larvae were transferred to fresh medium containing 3 mg/ml chloroquine (Sigma) as described (*Lőw et al., 2013*).

LysoTracker staining (GFP-Certified Lyso-ID red lysosomal detection kit; Enzo Life Sciences) of size-matched 90–96 hr fat bodies from fed and starved larvae was performed as previously described (*Rusten et al., 2004*; *Scott et al., 2004*). In brief, larvae were dissected in Schneider's *Drosophila* media (Gibco), inverted to expose fat bodies, and incubated in 100 µM LysoTracker Red DND-99 for 1 min. Inverted carcasses were then washed in 1X PBS and fat bodies were mounted onto a droplet of Vectashield (Vector Laboratories). Samples were imaged immediately on an inverted confocal microscope (LSM 510 Meta using 63×, NA 1.4 Plan Apochromat lens). Z-projections of three optical sections of fat body tissue, each 1 µm apart were used to quantify LysoTracker and Atg8a punctae in fat bodies using Imaris software (Bitplane). The number of punctate was quantified per fat body cell. Digital images for display were imported into Photoshop (Adobe) and adjusted for gain, contrast, and gamma settings.

Integrated intensities of Atg8a punctae in fat bodies were determined using Image J and normalized to Acn$^{WT}$. Integrated densities for pS437-Acinus and polyQ in eye discs were quantified using Image J software. Identical areas posterior to the morphogenetic furrow were quantified, thereby excluding dividing cells close to the furrow that were strongly stained for pS437-Acinus.

All immunofluorescence experiments were repeated at least three times with at least three samples each.

## Poly-Q dot blot filter retardation assay

Polyglutamine aggregates were detected with a modified filter assay (*Scherzinger et al., 1997*). Briefly, 25 heads from 2 week-old flies were homogenized in 200 µl Cytoplasmic Extraction Reagent I buffer and fractionated using NE-PER Nuclear and Cytoplasmic Extraction Reagents following the manufacturer's protocol (Thermo Fisher Scientific). Cytosolic fractions were adjusted to 1% SDS, incubated at room temperature for 15 min, denatured at 95°C for 5 min, and filtered through a 0.2 µm cellulose acetate membrane (Sterlitech Corporation) preequilibrated with 1% SDS. Membrane was washed twice with 0.2% SDS and blocked in TBS (100 mM Tris-HCl, pH 7.4, and 150 mM NaCl) containing 3% nonfat dried milk and probed with a mouse anti-Htt antibody (1:1,000; MAB5490; EMD Millipore). The bound antibodies are detected and quantified using anti-IR dye conjugated secondary antibodies and Odyssey scanner and software (LI-COR Biosciences).

## Statistical methods

Statistical significance was determined in Prism using log-rank for survival assays, chi square analysis for eye roughness frequencies, and one-way analysis of variance for multiple comparisons, followed by Tukey's test. To separate effects of treatment and genetic background we used two-way analysis of variance for multiple comparisons, followed by Bonferroni's test for individual comparisons. All bar graphs resulting from these analysis show means ±SD. For quantifications of fluorescence images and Western blots, at least three independent experiments were used. P values smaller than 0.05 are considered significant, and values are indicated with one (<0.05), two (<0.01), three (<0.001), or four (<0.0001) asterisks.

## Acknowledgements

We thank Drs. Karine Pozo and Qing Zhong for helpful comments to the manuscript and members of the Kramer lab for discussion and technical assistance. We thank Charles Tracy for performing the RT-qPCR assays. We thank Drs. Thomas Neufeld, Robin Hiesinger, Edward Giniger, Alysia Vrailas-Mortimer, Vienna Drosophila Resource Center, and the Bloomington Stock Center (NIH P40OD018537) for flies, Gàbor Juhàsz and the Developmental Studies Hybridoma Bank at The University of Iowa for antibodies and the Molecular and Cellular Imaging Facility at UT Southwestern Medical Center for help with electron microscopy. This work was funded by NIH grant EY010199 to HK and NSF Graduate Research Fellowship (4900835401–36068) to LKT.

## Additional information

### Funding

| Funder | Grant reference number | Author |
| --- | --- | --- |
| National Eye Institute | EY010199 | Helmut Krämer |
| National Science Foundation | 4900835401-36068 | Lauren K Tyra |

The funders had no role in study design, data collection and interpretation, or the decision to submit the work for publication.

### Author contributions

Nilay Nandi, Formal analysis, Investigation, Methodology, Writing—original draft, Writing—review and editing; Lauren K Tyra, Drew Stenesen, Investigation, Writing—review and editing; Helmut Krämer, Conceptualization, Supervision, Funding acquisition, Visualization, Writing—original draft, Project administration, Writing—review and editing

### Author ORCIDs

Nilay Nandi (iD) http://orcid.org/0000-0002-7088-4943
Helmut Krämer (iD) http://orcid.org/0000-0002-1167-2676

### Decision letter and Author response

Decision letter https://doi.org/10.7554/eLife.30760.031
Author response https://doi.org/10.7554/eLife.30760.032

## Additional files

### Supplementary files

• Supplementary file 1. Suppression of Acn-induced eye roughness by kinase knock down. The effect on eye morphology of GMR-Gal4-driven expression of UAS-RNAi transgenes with or without UAS-Acinus was scored and recorded as the number of flies with normal eyes (score = 1), eye with mild errors (score = 2), rough eyes (score = 3) or severely rough eyes (score = 4). The percentage suppression or enhancement was calculated from the average score for each RNAi transgene compared to UAS-Acn in the absence of RNAi transgenes. Positive or negative numbers indicate suppression and enhancement, respectively. Green or red colors highlight UAS-transgenes with more than 50% suppression or enhancement. Numbers in parenthesis indicated stock numbers of the Bloomington Drosophila stock center or, if starting with a 'V..', from the Vienna Drosophila Resource Center. All flies were raised at 28℃.

DOI: https://doi.org/10.7554/eLife.30760.025

• Supplementary file 2. Genetic interactions of Acn with p38b and Cdk5/p35 loss and gain of function. The effect on eye morphology of GMR-Gal4-driven expression of UAS-transgenes with or without UAS-Acinus was scored and recorded as the number of flies with normal eyes (score = 1), eye with mild errors (score = 2), rough eyes (score = 3) or severely rough eyes (score = 4). The percentage suppression or enhancement was calculated from the average score for each RNAi transgene

compared to UAS-Acn in the absence of RNAi transgenes. Positive or negative numbers indicate suppression and enhancement, respectively. Green or red colors highlight UAS-transgenes with more than 50% suppression or enhancement. Numbers in parenthesis indicated stock numbers of the Bloomington Drosophila stock center. All flies were raised at 28℃.

DOI: https://doi.org/10.7554/eLife.30760.026

• Supplementary file 3. Genotypes of flies used for each figure. Relevant genotypes are listed for all Figures.

DOI: https://doi.org/10.7554/eLife.30760.027

• Supplementary file 4. Key resources table.

DOI: https://doi.org/10.7554/eLife.30760.028

• Transparent reporting form

DOI: https://doi.org/10.7554/eLife.30760.029

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
