## [Decision Letter]

Thank you for submitting your article "Stress-Induced Cdk5 Activity Enhances Cytoprotective Basal Autophagy by Phosphorylating Acinus at Serine^437^" for consideration by *eLife*. Your article has been favorably evaluated by Ivan Dikic (Senior Editor) and three reviewers, one of whom is a member of our Board of Reviewing Editors. The reviewers have opted to remain anonymous.

The reviewers have discussed the reviews with one another and the Reviewing Editor has drafted this decision to help you prepare a revised submission.

Autophagy is a critical mediator of neuronal cell health. Ongoing basal clearance of aggregated or damaged proteins is associated with protection from neurodegeneration in both flies and humans. Previous studies from the authors identified a role for the Acinus (Acn) protein in promoting basal autophagy. In this manuscript, you provide evidence that Cdk5 genetically interacts with Acn to regulate basal level of autophagy, and that Cdk5 phosphorylates Acn at S437 to modulate its stability. The phospho-mimetic Acn^S437D^ promotes basal autophagy and rescues defects (e.g. reduced basal autophagy and shortened life span) of p35 mutants. You further demonstrate enhancement of the Cdk5/Acn-pS437 pathway in response to polyQ protein accumulation. Cdk5 was previously implicated in maintenance of neuronal health. This study identifies Acinus as a relevant target of Cdk5 activity as a compelling basis for this role. Overall, the findings are interesting and potentially important. The experiments are well executed and most conclusions are well supported by the data presented in this work. This manuscript will be suitable for publication in *eLife* after appropriate revisions.

Essential revisions:

1) In vitro kinase assays with purified proteins to show that Cdk5 kinase can directly phosphorylate Acinus.

2) The current data does not sufficiently support the conclusion that autophagy activity is induced by S437-Acn. The authors showed that the number of Atg8a puncta and LysoTracker-positive structures is increased in Acn^S437D^ larval fat bodies. The autolysosomes in Acn^S437D^ larval fat bodies are much larger and more abundant compared to Acn^WT^. The mean diameter of lysosomal/autolysosome structures is even increased more than 8-fold! Inhibition of autophagy at the step of autophagosome maturation and/or degradation results in the same phenotype. Does starvation-induced autophagy also cause accumulation of enlarged autolysosomes in fat bodies? Additional characterization of autophagy flux would strengthen the interpretations of autophagy induction induced by Acn-pS437. This could be achieved with assays using the tandem fluorescent Atg8 reporter and/or Atg8-II levels in animals fed BafilomycinA.

---

## [Author Response]

Essential revisions:1) in vitro kinase assays with purified proteins to show that Cdk5 kinase can directly phosphorylate Acinus.

We performed two sets of experiments to address this concern.

First, we prepared GST proteins fused to Acn^402-527^ (WT or S437A) and exposed them to phosphorylation with commercially available Cdk5/p35 complexes (Σ; 14-477). We observed Cdk5/p35-dependent S437-phosphorylation, that was not present in S437A controls (new Figure 6). Of note, we tried several other GST-Acn fusions for these experiments, but none was more stable against degradation than the GST-Acn-402-527 fusions.

Second, we over-expressed Strep-tag and Myc-tagged full-length Acinus proteins (WT or S437A) in S2 cells, immobilized and purified the proteins on Strep-Tactin beads before incubation with commercial Cdk5/p35. These experiments also showed Cdk5/p35-dependent S437-Acn phosphorylation (new Figure 6).

In combination, we believe these experiments convincingly address the reviewers’ concerns to show that Cdk5 kinase can directly phosphorylate Acinus at S437.

2) The current data does not sufficiently support the conclusion that autophagy activity is induced by S437-Acn. The authors showed that the number of Atg8a puncta and LysoTracker-positive structures is increased in Acn^S437D^ larval fat bodies. The autolysosomes in Acn^S437D^ larval fat bodies are much larger and more abundant compared to Acn^WT^. The mean diameter of lysosomal/autolysosome structures is even increased more than 8-fold! Inhibition of autophagy at the step of autophagosome maturation and/or degradation results in the same phenotype. Does starvation-induced autophagy also cause accumulation of enlarged autolysosomes in fat bodies?

Starvation-induced autophagy in fat bodies has been extensively studied by multiple labs. Examples include: (Scott et al., 2004) (Takats et al., 2014) (Takats et al., 2013). Neither of these papers quantified the changes in size from basal to starvations-induced autolysosomes. Importantly, however, all of these papers showed examples of electron micrographs from starved tissues containing large autolysosomes exceeding 1 µm or even 2 µm (e.g. Figure 2 in (Scott et al., 2004); Figure 2 in (Takats et al., 2013); or Figure 4 in (Takats et al., 2014).

Additional characterization of autophagy flux would strengthen the interpretations of autophagy induction induced by Acn-pS437. This could be achieved with assays using the tandem fluorescent Atg8 reporter and/or Atg8-II levels in animals fed BafilomycinA.

To address this concern, we performed experiments in which we blocked lysosomal acidification and degradation in larval fat bodies by feeding chloroquine to larvae following a well-established protocol to measure autophagic flux in larval fat bodies (Low et al., 2013; Mauvezin et al., 2014; Lorincz et al., 2017). When staining these chloroquine-treated fat bodies for endogenous ATG8, we observed significant increases in ATG8a punctae and ATG8a signal intensity for wild-type starved or for fed S437D phospho-mimetic mutants consistent with increased autophagic flux (new Figure 2 and Figure 2—figure supplement 1).

To keep this figure manageable, we have split the previous Figure 2 and separated the analysis of the effects of S437A and S437D mutants on starvation-resistance and longevity into the new Figure 3.

In addition, we like to point out that the beneficial effects endowed by the S437D phospho-mimetic mutation on starvation-resistance (new Figure 3), longevity (new Figure 3, Figure 7), and suppression of polyQ protein accumulation (new Figure 8) would be hard to explain as a consequence of defects in lysosomal degradation or reduced autophagosome maturation. By contrast, these effects are entirely consistent with beneficial effects of increased autophagy in the context of longevity and proteinopathies previously reported by several labs (some of many examples: (Sarkar et al., 2007; Simonsen et al., 2008; Nandi et al., 2014; Menzies et al., 2015; Gelino et al., 2016). We have now addressed this issue in the first paragraph of the Discussion.